# A photocatalytic redox cycle over a polyimide catalyst drives efficient solar-to-H$_2$O$_2$ conversion

Wenwen Chi[1,4], Yuming Dong [1,4] ✉, Bing Liu[1], Chengsi Pan [1], Jiawei Zhang[1], Hui Zhao[1], Yongfa Zhu [2] ✉ & Zeyu Liu[3]

Circumventing the conventional two-electron oxygen reduction pathway remains a great problem in enhancing the efficiency of H$_2$O$_2$ photosynthesis. A promising approach to achieve outstanding photocatalytic activity involves the utilization of redox intermediates. Here, we engineer a polyimide aerogel photocatalyst with photoreductive carbonyl groups for non-sacrificial H$_2$O$_2$ production. Under photoexcitation, carbonyl groups on the photocatalyst surface are reduced, forming an anion radical intermediate. The produced intermediate is oxidized by O$_2$ to produce H$_2$O$_2$ and subsequently restores the carbonyl group. The high catalytic efficiency is ascribed to a photocatalytic redox cycle mediated by the radical anion, which not only promotes oxygen adsorption but also lowers the energy barrier of O$_2$ reduction reaction for H$_2$O$_2$ generation. An apparent quantum yield of 14.28% at 420 ± 10 nm with a solar-to-chemical conversion efficiency of 0.92% is achieved. Moreover, we demonstrate that a mere 0.5 m$^2$ self-supported polyimide aerogel exposed to natural sunlight for 6 h yields significant H$_2$O$_2$ production of 34.3 mmol m$^{-2}$.

Hydrogen peroxide (H$_2$O$_2$), which serves as an eco-friendly oxidant and a versatile energy carrier, has widespread applications in healthcare, disinfection, wastewater treatment, and chemical synthesis[1]. The global market demand for H$_2$O$_2$ is estimated to possibly increase at a compound annual growth rate of 4.6%, reaching 5.7 million tons by 2028[2]. Currently, more than 95% of the H$_2$O$_2$ in the market is predominantly produced through the anthraquinone process, which relies on noble palladium-based catalysts and consumes substantial energy resources. Moreover, the conventional homogeneous anthraquinone process has been criticized for extracting produced H$_2$O$_2$ and generating toxic byproducts[3]. Consequently, exploration of environmentally friendly approaches for the synthesis of H$_2$O$_2$ is desirable.

Artificial photosynthesis of H$_2$O$_2$ using organic semiconductors represents an advanced approach toward a sustainable and environmentally friendly future[2,4]. To date, powder photocatalysts that can harness water, oxygen, and sunlight to generate H$_2$O$_2$ have been reported[5]. For example, g-C$_3$N$_4$ derivatives, resorcinol-formaldehyde resins, conjugated polymers, covalent triazine frameworks, and covalent organic frameworks have exhibited H$_2$O$_2$ production[6]. However, previously established photocatalytic systems have significant limitations due to the direct collision of photogenerated electrons with oxygen, which results in rapid charge carrier recombination and diverse photoreduction pathways[7–9]. Given the sluggish nature of the two-electron reduction of O$_2$, photoinduced electrons are prone to recombine with holes before reacting with dissolved oxygen in water, significantly decreasing the quantum efficiency. In addition, the O$_2$ reduction process involves two competing reactions, i.e., one-electron reduction and four-electron reduction of O$_2$, severely limiting the selectivity for H$_2$O$_2$. These issues result in insufficient activity for photocatalytic H$_2$O$_2$, which is far from sufficient for large-scale

[1]International Joint Research Center for Photoresponsive Molecules and Materials, Key Laboratory of Synthetic and Biological Colloids, School of Chemical and Material Engineering, Jiangnan University, Wuxi, China. [2]Department of Chemistry, Tsinghua University, Beijing, China. [3]School of Environmental and Chemical Engineering, Jiangsu University of Science and Technology, Zhenjiang, China. [4]These authors contributed equally: Wenwen Chi, Yuming Dong. ✉ e-mail: dongym@jiangnan.edu.cn; zhuyf@tsinghua.edu.cn

production. Hence, the development of an innovative reaction pathway for the photosynthesis of $H_2O_2$ is imperative.

In green plants, photogenerated electrons are effectively channeled through redox species such as plastoquinone and NADP, which is advantageous for preventing electron-hole recombination and enhancing quantum efficiency[10,11]. The utilization of sufficiently reducing species as reaction intermediates to drive the reduction of $O_2$ is an ideal approach for the photosynthesis of $H_2O_2$. Inspired by the high activity and selectivity advantages of the industrial anthraquinone process, polymer photocatalysts incorporating anthraquinone units as reductive sites or cocatalysts have been employed to promote two-electron oxygen reduction for $H_2O_2$ synthesis[12–14]. Nonetheless, indepth insights into the anthraquinone photocatalytic mechanism for solar-driven $H_2O_2$ production are lacking. Polyimides with anthraquinone-like structures have been used as photoreceptors or electronic materials due to their combination of photostability, electronic structure tuning, and redox properties[15,16]. These qualities align well with the requirements of the photocatalytic system that we want to design. In particular, anion radicals of produced aromatic imides and diimides have been identified as potent electron reductants[17–20]. However, a conceptually related mechanism involving photoinduced anion radical intermediates has not yet been comprehensively revealed for $H_2O_2$ photosynthesis.

In this work, we developed a covalently crosslinked polyimide aerogel photocatalyst featuring a reductive C = O group, designated PI-BD-TPB. A notable $H_2O_2$ concentration of 2.85 mM $h^{-1}$ coupled with an apparent quantum yield (AQY) of 14.28% at 420 ± 10 nm was attained.

Impressively, even a mere 0.5 $m^2$ self-supported polyimide aerogel exposed to natural sunlight displayed a remarkable $H_2O_2$ yield of 34.3 mmol $m^{-2}$, rendering it conducive to separation and recycling in large-scale applications. More importantly, we revealed the pathway of an intermediate-induced photocatalytic redox cycle for $H_2O_2$ synthesis. Under photoexcitation, carbonyl groups on the polyimide aerogel surface were reduced to anion radicals, which were oxidized by $O_2$ to generate $H_2O_2$, followed by reversion to carbonyl groups. Through comprehensive in situ experimental investigations along with theoretical calculations, we deeply demonstrated that the redox cycle mechanism enhanced $O_2$ adsorption and lowered the energy barrier of the $O_2$ reduction reaction, thereby significantly boosting the overall photosynthetic production of $H_2O_2$.

## Results and discussion
### Photocatalytic performance and scalable production of $H_2O_2$
The PI-BD-TPB aerogel photocatalyst for solar-driven $H_2O_2$ production was prepared via condensation of the triangular aromatic triamine 1,3,5-tris[4-amino(1,1-biphenyl-4-yl)]-benzene (TPB) as the donor unit and the linear 3,3′,4,4′-biphenyltetracarboxylic dianhydride (BD) as the acceptor unit by the sol-gel-thermal imidization route (Supplementary Figs. 3, 4). The photocatalytic performance of PI-BD-TPB for non-sacrificial $H_2O_2$ production was assessed under simulated sunlight illumination. The average $H_2O_2$ concentration for PI-BD-TPB in water under a saturated $O_2$ atmosphere was 2833 μM $h^{-1}$, which largely exceeded that of most reported polymeric photocatalysts (Fig. 1a). The $H_2O_2$ yield in $O_2$-saturated pure water was 3.78 times greater than that

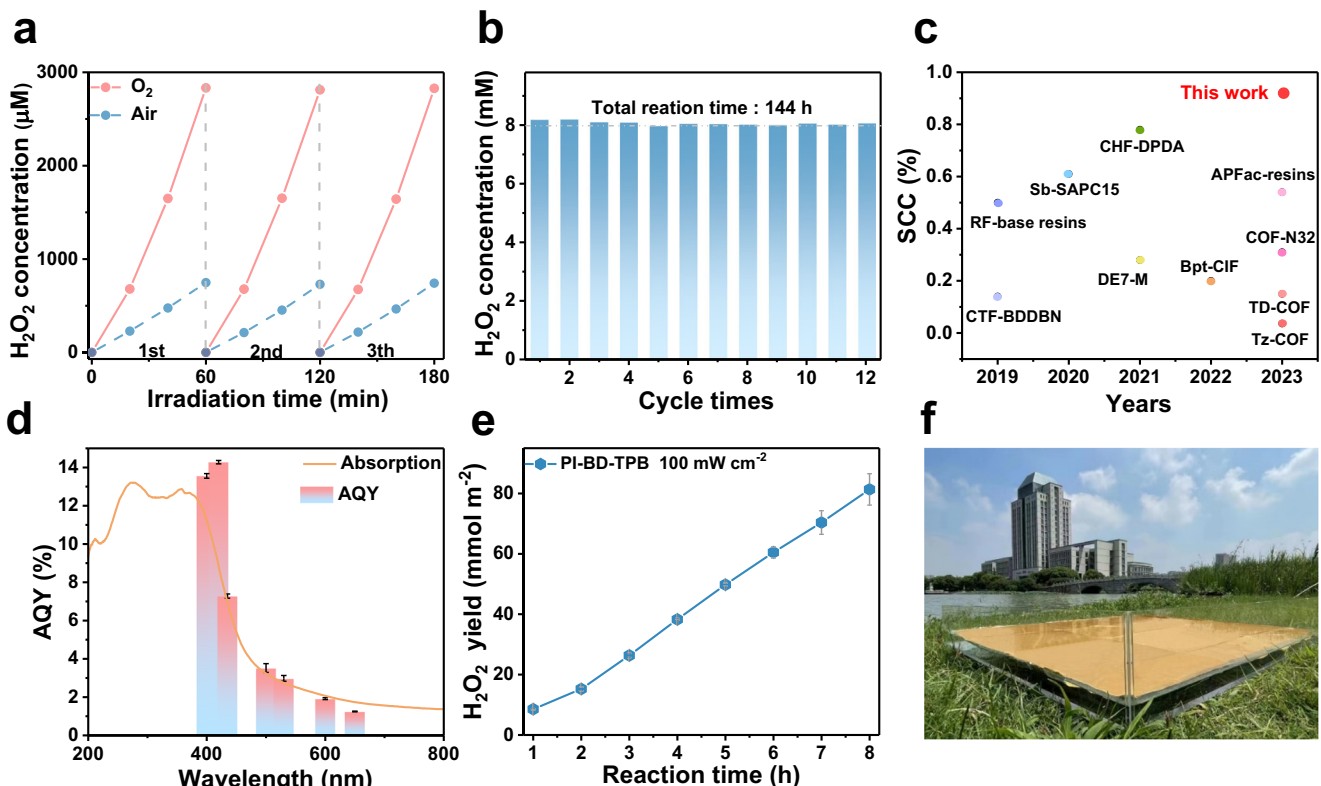

**Fig. 1 | Photocatalytic performance and scalable production towards $H_2O_2$ production of the PI-BD-TPB aerogel photocatalyst. a** The stable $H_2O_2$ production with $O_2$ or air atmosphere of the PI-BD-TPB aerogel under AM 1.5 illumination. Reaction conditions: 15 mg aerogel photocatalyst, 20 mL $H_2O$, 298 K. **b** The stable cyclic photocatalytic $H_2O_2$ production of the PI-BD-TPB aerogel. **c** SCC efficiency comparison of the PI-BD-TPB aerogel with other reported photocatalysts, See Supplementary Table S1 for more detailed information including references. **d** The wavelength-dependent apparent quantum yield (AQY) of the PI-BD-TPB aerogel in the photocatalytic $H_2O_2$ production, 400 ± 10 nm, 420 ± 10 nm, 435 ± 10 nm, 500 ± 10 nm, 530 ± 10 nm, 600 ± 10 nm, and 650 ± 10 nm, Error bars on mean values are standard deviations of three independent $H_2O_2$ production tests. **e** Practical performance test based on the PI-BD-TPB aerogel under irradiation intensity of 100 mW $cm^{-2}$. **f** Digital image of the photocatalytic $H_2O_2$ production under natural sunlight on Jiangnan University campus using the PI-BD-TPB aerogel membrane from a side view with the 0.5 $m^2$ panel reactor containing 10 L $H_2O$.

in air-saturated pure water (748.47 $\mu M\,h^{-1}$), strongly indicating the indispensable role of oxygen in the reaction system. Nevertheless, a remarkable accumulation amount was exhibited, up to 14.34 mM $H_2O_2$ after 12 h (Supplementary Figs. 8). In addition to exhibiting high activity, PI-BD-TPB displayed ultradurable performance for a continuous 144 h photocatalytic run, demonstrating satisfactory photostability (Fig. 1b). According to Fourier transform infrared (FTIR) spectroscopy, solid-state $^{13}C$ NMR spectroscopy, X-ray diffraction (XRD), scanning electron microscopy (SEM), transmission electron microscopy (TEM) and UV-vis absorption results, the PI-BD-TPB aerogel exhibited no significant structural, morphological or absorption edge changes after long-term reaction (Supplementary Figs. 9–13).

The solar-to-chemical conversion efficiency (SCC) of PI-BD-TPB was measured to be 0.92%, surpassing the typical plant efficiency (0.10%) (Fig. 1c and Supplementary Fig. 14). As shown in Fig. 1d, the AQY of PI-BD-TPB exhibited a wavelength dependence, which was in accordance with its absorption spectrum. The maximum AQY of PI-BD-TPB at $420 \pm 10$ nm reached 14.28%. Moreover, PI-BD-TPB exhibited negligible activity for $H_2O_2$ decomposition under continuous irradiation for 24 h (Supplementary Fig. 15). The macroscopic polyimide aerogel photocatalyst not only has good photostability and photocatalytic reactivity, but also is convenient for separation and recyclability. Benefiting from the facile process, macroscopic PI-BD-TPB aerogel with membrane shapes was also prepared, confirming the feasibility of large-scale fabrication. We performed an activity test under $100\,mW\,cm^{-2}$ irradiation for 8 h, and PI-BD-TPB exhibited a remarkable $H_2O_2$ yield of 88.6 mmol $m^2$ (Fig. 1e). Motivated by the excellent photocatalytic performance of PI-BD-TPB, a scalable test for photocatalytic $H_2O_2$ production in an outdoor environment was conducted under natural sunlight as the energy source. A 0.5 $m^2$ polyimide aerogel membrane photocatalyst effectively produced a $H_2O_2$ yield of 34.3 mmol $m^2$, demonstrating its practical value for large-scale hydrogen peroxide production (Fig. 1f and Supplementary Figs. 16–18).

## Carbonyl group photo-reduction to anion radical intermediate

To understand the excellent photocatalytic performance in $H_2O_2$ production, we investigated the structural characteristics of the PI-BD-TPB aerogel. We successfully fabricated covalently crosslinked polyimide aerogels with π-conjugated and π-stacked donor-acceptor structures. PI-BD-TPB, which had a low density (ca. 37.78 mg/cm³), was so light that it could rest on top of a dandelion (Fig. 2a). FTIR spectroscopy was employed to monitor the chemical structure (Fig. 2b). Bands were clearly observed in the PI-BD-TPB curve at 1777.5, 1708.9 and 738.6 $cm^{-1}$, representing the asymmetric stretching, symmetric stretching and bending vibrations of carbonyl groups, respectively. The band at 1358.1 $cm^{-1}$ was assigned to the stretching vibration of -C − N, indicating the complete condensation of BD and TPB monomers[21]. Solid-state $^{13}C$ NMR spectroscopy (Fig. 2c) was used to further confirm the formation of the polyimide aerogel. The observed signal at 166.5 ppm indicated the presence of a carbonyl carbon atom in the imide ring. Additional chemical shifts in the range of 100–150 ppm were attributed to phenyl carbon atoms[22]. Moreover, PI-BD-TPB showed a new band located at 288.5 eV in the high-resolution C 1s spectrum, corresponding to the chemical bond of -C = O (Supplementary Fig. 19)[23]. Collectively, the FTIR, solid-state $^{13}C$ NMR and XPS spectra demonstrated that the prepared PI-BD-TPB aerogel photocatalyst is rich in carbonyl groups.

Additional characterizations were carefully performed to understand the features of the PI-BD-TPB aerogel. PAA is the precursor of polyimide, whose molecular weight distribution determines the molecular weight of the final product. Due to the insolubility of PI-BD-TPB, we measured the molecular weight of the PAA gel powders via gel permeation chromatography in NMP. The average molecular weight (Mw) and corresponding polydispersity index (PDI, PDI = Mw/Mn, Mn = 3.4 kDa) were 8.0 kDa and 2.35 (Supplementary Fig. 20). SEM and

TEM images of the PI-BD-TPB aerogel (Supplementary Figs. 21–23) revealed cross-linked spherical particles with an average diameter of ~1 μm. High-resolution transmission electron microscopy (HRTEM) images revealed that PI-BD-TPB was locally crystalline in nature. The interlayer distance was measured to be 0.43 nm (Supplementary Fig. 24). Furthermore, the XRD peak of PI-BD-TPB was located at $2\theta \approx 20.3°$, representing the 0.43 nm interlamellar d-spacing of π–π stacking (Supplementary Fig. 25). The porosity of PI-BD-TPB was assessed using $N_2$ sorption measurements at 77.3 K, and the Brunauer–Emmett–Teller surface area was calculated to be 372.8 $m^2\,g^{-1}$. By employing a nonlocal density functional theory model, its pore size was determined to be ~1.5 nm (Supplementary Fig. 26). The macroscopic pore size distribution on the surface of the BD-TPB aerogel was analyzed by mercury intrusion porosimetry (Supplementary Fig. 27). The macroscopic pore size on the surface of BD-TPB was ~2 μm.

Thermogravimetric analysis revealed that the PI-BD-TPB aerogel had a high thermal stability up to 550 °C (Supplementary Fig. 28). The chemical stability of the PI-BD-TPB aerogel was investigated by immersing it in diverse solvents. Notably, the FTIR spectra of PI-BD-TPB after soaking in different solvents were almost unchanged, confirming its excellent chemical stability (Supplementary Fig. 29). The good chemical and thermal stability are ascribed to the strong imide linkage and highly conjugated structure. The average ζ potential of PI-BD-TPB was − 55.1 mV in $H_2O$ aqueous solution, indicating strong adsorption of $H^+$ (Supplementary Fig. 30). PI-BD-TPB displayed a hydrophilic surface with a contact angle of 40.4° (Supplementary Fig. 31). The macroscopic polyimide aerogel was cylindrical, with a diameter of 3 cm and a height of 2 cm. Compared to powder with the same weight, it was characterized by a low density (ca. 37.78 mg/cm³). (Supplementary Figs. 32–34).

Adsorption is a special feature of aerogel materials. The 0.53 g PI-BD-TPB aerogel rapidly absorbed 13.89 g of $H_2O$ after 5 min, which was approximately 25 times its weight, exhibiting an effective adsorption capacity (Supplementary Figs. 35, 36). In addition, the PI-BD-TPB aerogel with excellent mechanical behavior could hold 200 times its own weight (Supplementary Fig. 37). A compression−recovery test was performed to estimate the mechanical durability, and the PI-BD-TPB aerogel, as an elastic material, exhibited excellent reversible compressibility at strains of 10%, 20%, 30%, 40%, 50% and 60% (Supplementary Fig. 38). Benefiting from the facile synthetic route, a macroscopic polyimide membrane was also prepared, further exhibiting high operability (Supplementary Fig. 39). In summary, the covalently crosslinked polyimide aerogel has a low density, hydrophilicity, an effective absorption ability, high chemical and thermal stability, excellent resilience and good mechanical behavior.

Theoretical calculations showed that the highest occupied molecular orbital (HOMO) was mainly distributed on the TPB unit, while the lowest unoccupied molecular orbital (LUMO) was distributed on the BD unit, suggesting transfer of photogenerated electrons from the TPB unit to the BD unit (Supplementary Fig. 40). The -C = O groups on BD-TPB had electron affinity characteristics according to the electrostatic potential distribution (Supplementary Fig. 41). The electrostatic potential of PI-BD-TPB under photoexcitation indicated that the carbonyl group had a strong ability to extract electrons (Supplementary Fig. 42). Meanwhile, the related photophysical experiments containing electrochemical impedance spectroscopic spectra, photocurrent spectra, surface photovoltage spectra and transient fluorescence spectra were performed. We confirm that PI-BD-TPB aerogel with donor-acceptor structure has excellent charge separation efficiency and extended carrier lifetime. (Supplementary Fig. 46–49).

To investigate the reductive property of -C = O in PI-BD-TPB, we performed cyclic voltammetry measurements (Supplementary Fig. 43). Two pairs of peaks were observed at approximately 0.509 V / 1.674 V and 0.009 V / 0.905 V (vs. RHE), which could be assigned to enolization of carbonyl oxygen, indicating that the R-C = O in the imide

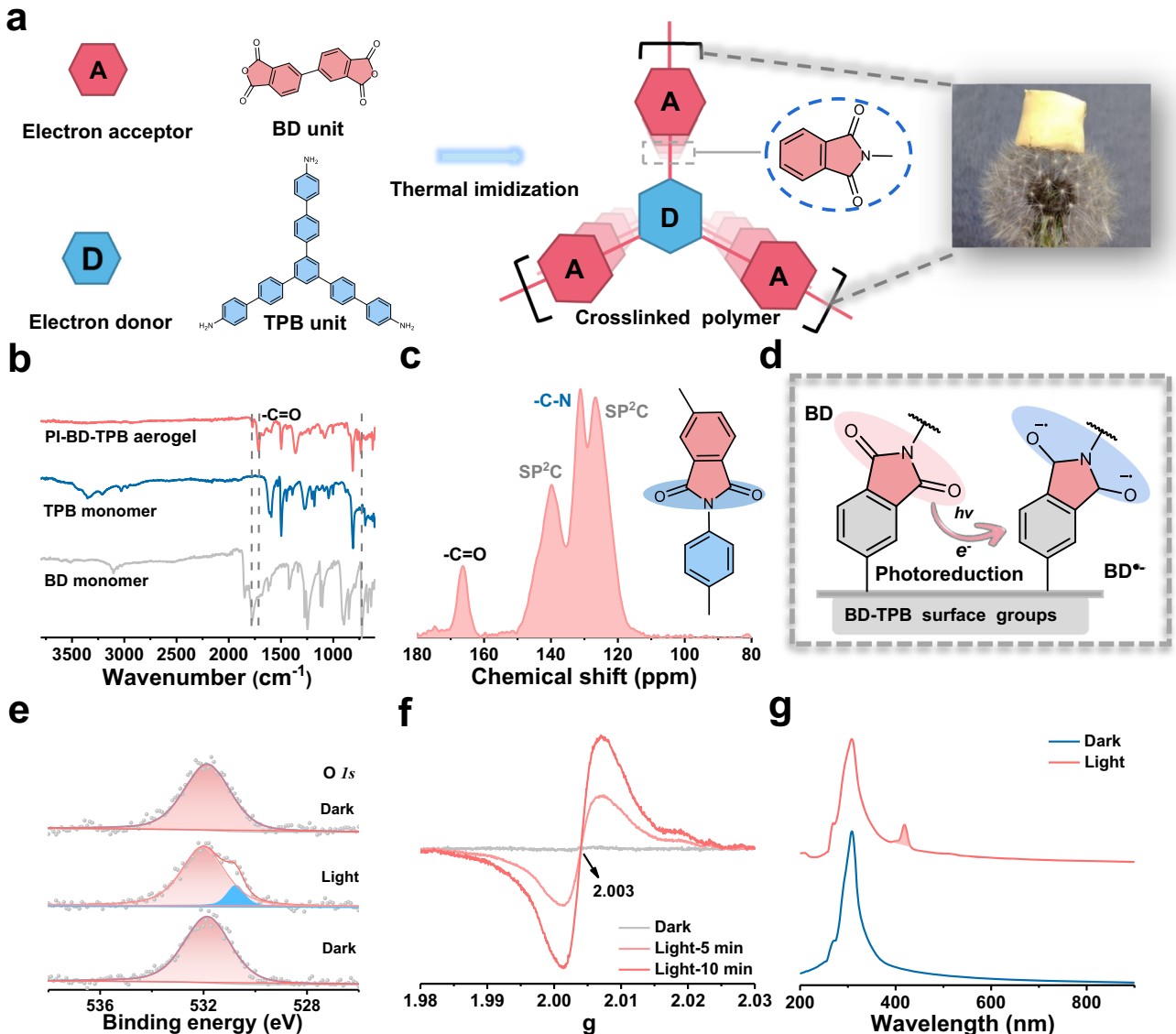

**Fig. 2 | PI-BD-TPB with C = O group and the photoreduction generation of anion radical intermediate. a** Schematic diagram of synthesis of the PI-BD-TPB aerogel with π-conjugated and π-stacked D-A structure. **b** FTIR spectra of the monomer BD and TPB, with the PI-BD-TPB aerogel. **c** Solid-state $^{13}$C NMR spectra of the PI-BD-TPB aerogel. **d** Schematic diagram of anion radical intermediate formation via photogenerated electron transfer and reduction. **e** In situ X-ray photoelectron spectroscopy spectrum of O $1s$ on the PI-BD-TPB aerogel. **f** Electron paramagnetic resonance spectra of the PI-BD-TPB before and after light irradiation (300 W Xe lamp). **g** Ultraviolet-visible absorption spectra of BD-TPB and BD-TPB* anion radical in DMF solution with $Na_2S_2O_4$ as electron donor under Ar atmosphere.

ring stores electrons[24,25]. Mott–Schottky plots, X-ray photoelectron spectroscopy valence band spectra, and ultraviolet-visible absorption spectra were obtained to determine the band gap and band position of PI-BD-TPB (Supplementary Figs. 44, 45). The band gap of PI-BD-TPB was estimated to be 2.21 eV. The conduction band edge ($E_{CB}$) of PI-BD-TPB was calculated to be − 0.19 V (*vs*. RHE), and the valence band edge ($E_{VB}$) was 2.02 V (*vs*. RHE). These results confirmed that the PI-BD-TPB aerogel provided a thermodynamically favorable driving force for the reduction of R-C = O and two-electron oxidation of $H_2O$. Note that the anion radical is highly likely to form on the surface of PI-BD-TPB during photoexcitation due to the hyper-conjugated structure (Fig. 2d).

The structural variation of the C = O group in PI-BD-TPB was monitored by in situ X-ray photoelectron spectroscopy (Fig. 2e). Initially, in the dark, the O $1s$ XPS peak was resolved into one major component (532.1 eV) for the C = O group. Under light irradiation, a prominent new peak at 530.8 eV attributed to the C − O bond emerged[26,27]. This observation indicated that the C = O group in the imide ring underwent structural transformation, in which it received

photogenerated electrons to form a C − O group. In contrast, the C = O signal decreased. After the light was turned off, the O $1s$ peak reverted to the original state, proving the transformation of C = O bonds to C − O bonds. To gain further insight into the structural variation of the C = O group, electron paramagnetic resonance (EPR) spectroscopy was performed on PI-BD-TPB (Fig. 2f). There was no obvious signal at $g$ = 2.003 in the dark. Upon light irradiation, the intensity of the signal at $g$ = 2.003 significantly increased, indicating that photoinduced electrons were transferred to carbonyl groups, corresponding to imide radical formation[28]. These changes demonstrated that the reduction of R-C = O to the anion radical occurred on the PI-BD-TPB surface during the photocatalytic process. Moreover, the presence of superoxide radicals was excluded based on the EPR measurement results (Supplementary Fig. 50). The above results confirmed that the R-C = O groups in the photocatalyst structure were reduced by photogenerated electrons, forming anion radicals in our system.

UV–vis spectroscopy and fluorescence tests were conducted to probe the formation of anion radicals. In the UV–vis absorption

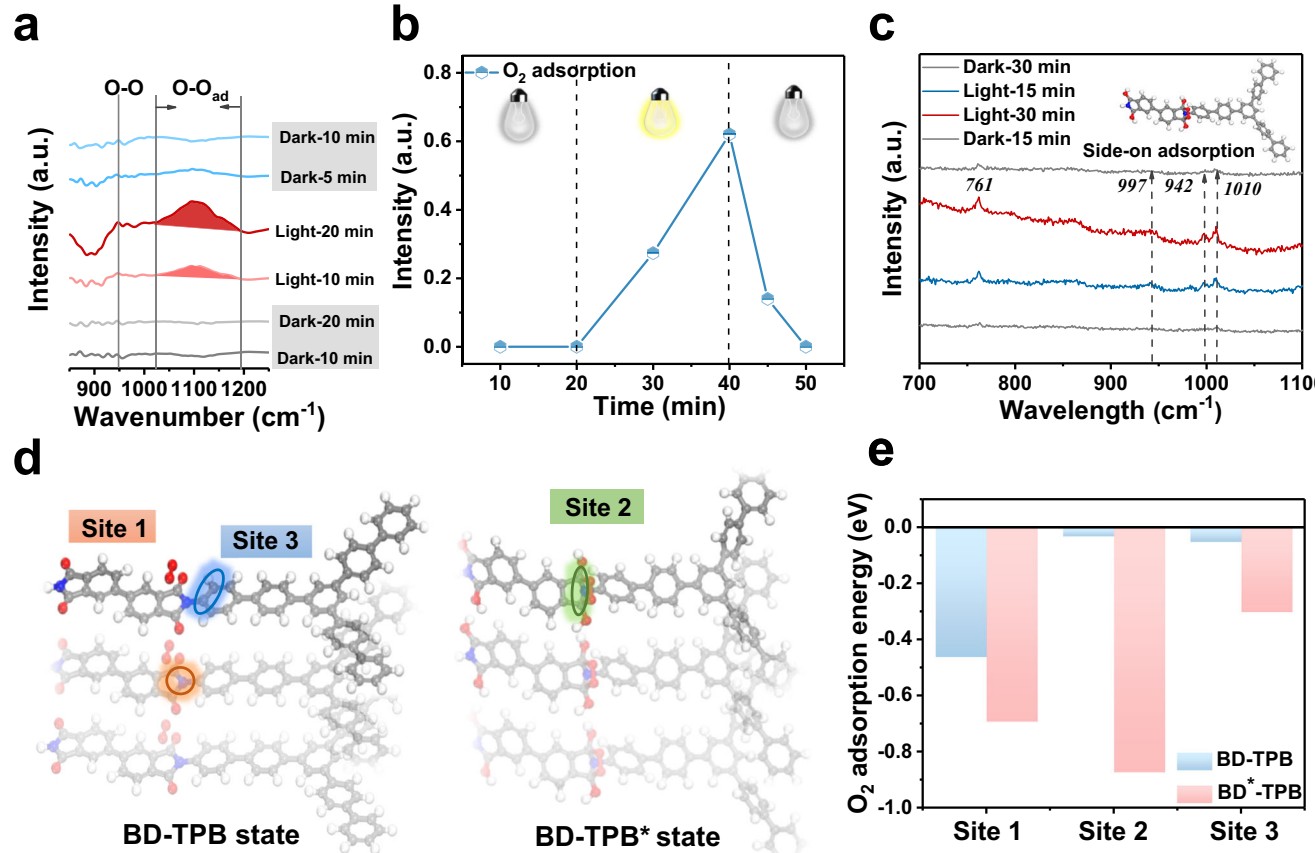

**Fig. 3 | Enhanced oxygen adsorption by photoreduction formed anion radical intermediate. a** Experimental in situ FTIR spectra and (**b**) The peak intensity of $O_2$ adsorption recorded during photoreaction in a 2-propanol aqueous solution with saturated oxygen (2-propanol as electron donor, 10% v/v). **c** In situ Raman spectra of the PI-BD-TPB aerogel recorded during photoreaction in a 2-propanol aqueous solution with saturated oxygen, a 785 nm laser was taken as light source to shine directly on the photocatalyst surface, and a computer synchronously collected the Raman signals. **d** Different adsorption site a of $O_2$ on -C = O state (BD-TPB) and radical anion state (BD-TPB*) respectively. **e** The adsorption energy of $O_2$ on different sites of C = O state (BD-TPB) and radical anion state (BD-TPB*), respectively.

spectrum (Fig. 2g), the 250–300 nm absorption peak was from the carbonyl group on the imide ring. The carbonyl group received electrons provided by $Na_2S_2O_4$ as the electron donor upon light illumination[29], and the structure changed; thus, the absorption appeared to be red-shifted. The new peak at 418 nm corresponded to the carbon–oxygen anion radical formed by the carbonyl group that obtained electrons[30,31]. Meanwhile, we performed the fluorescence test of BD-TPB under dark and light conditions[31,32]. Under dark conditions, PI-BD-TPB exhibited an obvious signal at 460 nm. The signal intensity at 460 nm decreased after light irradiation, indicating that the carbonyl group received photogenerated electrons and formed the anion radical, leading to a decrease in the fluorescence signal (Supplementary Fig. 51). Thus, the reduction of the -C = O group in our system generates the anion radical under light irradiation, which is consistent with the experimental results. We also confirmed that the photogenerated holes in the PI-BD-TPB aerogel could be consumed via a direct two-electron water oxidation pathway for $H_2O_2$ generation on the C atom in the TPB donor unit (Supplementary Figs. 52–56). Therefore, the photogenerated electrons and holes in PI-BD-TPB were used for the reduction of the -C = O group and oxidation of $H_2O$, respectively.

### Enhanced oxygen adsorption by anion radical intermediate

To thoroughly investigate the role of the anion radical intermediate in the oxygen reduction reaction, we conducted in situ FTIR spectroscopy under operando conditions. The entire process was carried out in a 2-propanol solution with a saturated $O_2$ atmosphere. As shown in

Fig. 3a, b, no oxygen adsorption signal was observed under dark conditions, while the characteristic peak for the O − O stretching signal at 948.22 $cm^{-1}$ and the O − O adsorption signal at 1023.7 − 1196.7 $cm^{-1}$ were detected during the illumination process[33,34]. In particular, a weak O − O adsorption signal could still be detected 5 min after the light was turned off, indicating that the anion radical intermediate significantly enhanced oxygen adsorption. In situ Raman spectra were also obtained to provide direct evidence (the test conditions were the same as those in the in situ FTIR spectroscopy measurements). The 761.2 $cm^{-1}$ band was assigned to the in-plane bending mode of the imide moiety[35]. No signal was detected in the dark, indicating that the $O_2$ adsorption of the PI-BD-TPB aerogel was weak. However, a new broad band appeared at 942 $cm^{-1}$ during the irradiation process, corresponding to the O − O stretching of the imide ring (Fig. 3c)[36,37]. These findings, which align with the results of in situ FTIR spectroscopy measurements, strongly support that the anion radical intermediate promotes $O_2$ adsorption.

The adsorption of oxygen molecules on photocatalysts has been recognized as the most critical step of the $O_2$ reduction process[38]. The $O_2$ adsorption property of the anion radical intermediate was investigated by first-principles calculations to understand the correlation between the structure and performance. We calculated the adsorption of $O_2$ in the -C = O state (BD-TPB) and radical anion state (BD-TPB*). Initially, the oxygen adsorption on different sites in the initial PI-BD-TPB state was weak. In contrast, the oxygen adsorption energy on each site was significantly decreased in the protonated anion intermediate state (Fig. 3d, e). In particular, the two symmetrical carbon atoms on

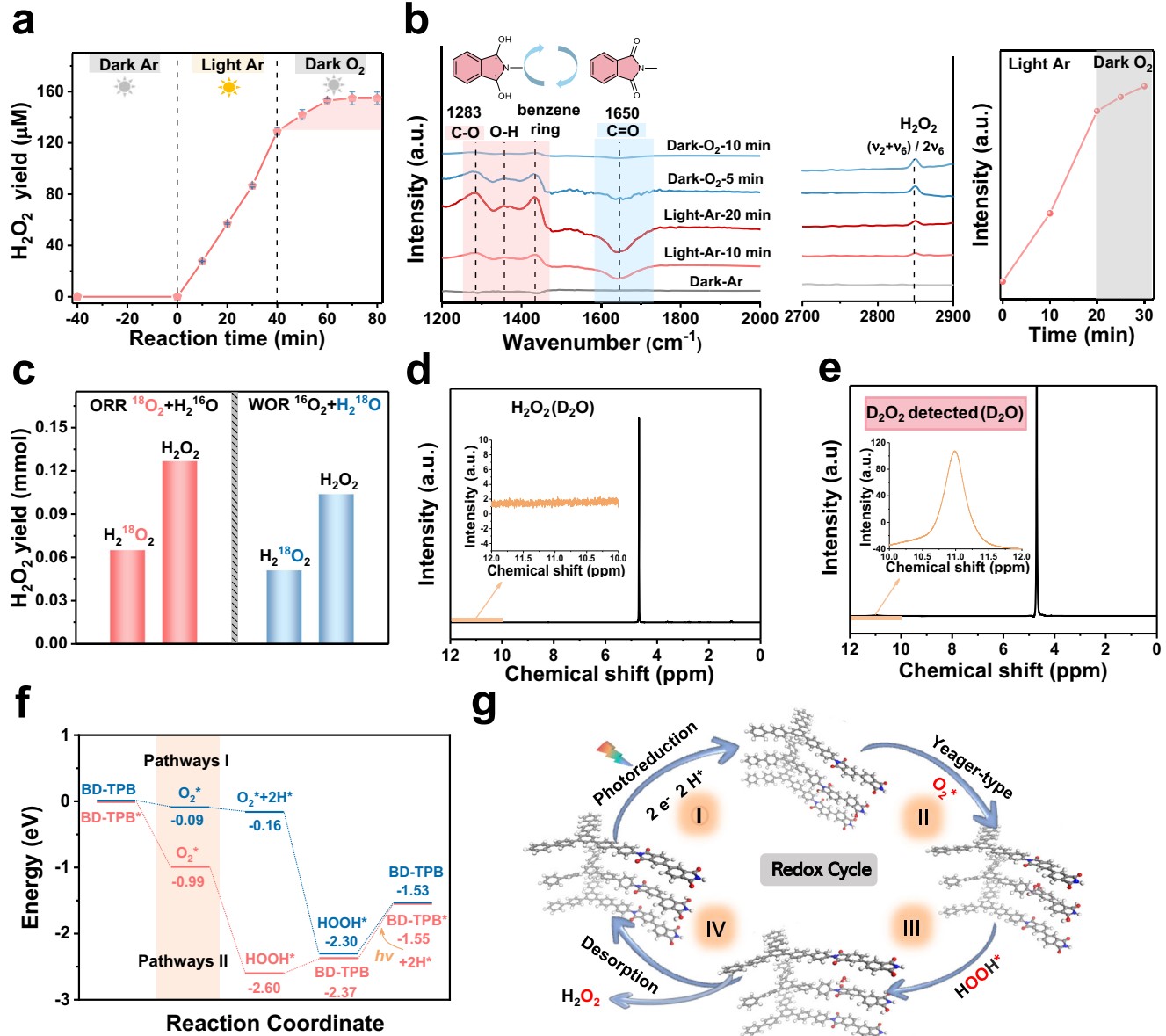

**Fig. 4 | Redox cycle mechanism mediated by photoinduced anion radical intermediate. a** $H_2O_2$ production on the PI-BD-TPB aerogel under different control conditions, Error bars on mean values are standard deviations of three independent tests, reaction conditions: catalyst (50 mg), water (25 mL), Ar atmosphere in the light or $O_2$ atmosphere in the dark. **b** In situ FTIR spectrum of the PI-BD-TPB aerogel with Ar atmosphere under 300 W Xe lamp and $O_2$ gas purging in the dark at 1200–2000 $cm^{-1}$ and 2700–2900 $cm^{-1}$, the peak intensity of $H_2O_2$ species on "Light-

Ar /Dark-$O_2$" condition. **c** Isotopic experiments with $^{18}O_2$ or $H_2^{18}O$ for $H_2O_2$ production on the PI-BD-TPB photocatalyst, See Supplementary Table 2 for more detailed information. **d** $^1H$ spectrum of $H_2O_2$ in $D_2O$. **e** $^1H$ spectrum of the photocatalytic $D_2O_2$ production of the BD-TPB in $D_2O$. **f** Calculated free energy diagrams of $H_2O_2$ production from $O_2$ reduction pathway by model systems of C = O state (BD-TPB) and radical anions state (BD-TPB*), respectively. **g** Key steps of $H_2O_2$ production from $O_2$ reduction pathway on the PI-BD-TPB aerogel photocatalyst.

the imide ring (site 2) were found to be particularly favorable for the Yeager-type (side-on) $O_2$ adsorption, which was consistent with the results of the experimental in situ FTIR and Raman spectra. According to the literature, the Yeager-type (side-on) adsorption with enhanced adsorption affinity for $O_2$ is extremely conducive to the oxygen reduction reaction for $H_2O_2$ synthesis[34].

### Redox cycle mechanism via anion radical intermediate

To elucidate the $O_2$ reduction pathway for $H_2O_2$ production, we devised a continuous "Light-Ar/Dark-$O_2$" experiment similar to plant photosynthesis (Fig. 4a)[39]. The system was purged with Ar gas to exhaust $O_2$ gas in the pure water before turning on the light. Approximately 129 μM $H_2O_2$ was generated by PI-BD-TPB under the "Light-Ar" condition, showing that the photogenerated holes oxidized

water to synthesize $H_2O_2$. Significantly, when the light was turned off and the system was subsequently purged with $O_2$ gas, approximately 23 μM $H_2O_2$ continued to be detected, revealing that the formed intermediate could combine with $O_2$ to produce $H_2O_2$ within a certain time. This result implied the importance of the radical intermediate in the dark-driven reaction for $H_2O_2$ production. Multiple experiments verified that the amount of $H_2O_2$ continuously increased within 10 min under the "Dark-$O_2$" condition. As a result, we speculate that the carbonyl group on the catalyst surface stored photogenerated electrons during the photoreaction and formed the anion intermediate. In the dark, the anion intermediates released the stored electrons and directly reduced $O_2$ to $H_2O_2$. The $O_2$ reduction reaction pathways for $H_2O_2$ generation via the redox intermediate greatly improved the utilization rate of photogenerated electrons.

In situ FTIR spectra confirmed the above hypothesis. In situ FTIR spectra of PI-BD-TPB were continuously recorded during the light / dark reaction to verify the structural variation of the PI-BD-TPB photocatalyst. As shown in Fig. 4b, the signal intensity of the $C = O$ groups at 1650 cm$^{-1}$ decreased under the light-Ar condition and reverted to the original state under the dark-$O_2$ condition, and the formation and disappearance of the peak at 1283 cm$^{-1}$ was attributed to the $C - O$ group[40]. The peak at ~1354 cm$^{-1}$ corresponding to $O - H$ bending displayed a similar trend to that of the $C - O$ group[41], indicating that the photogenerated electrons reduced the carbonyl group to the protonated anion intermediate. This observation aligns with the prior results of zeta potential experiments. The new infrared vibration signal at 2849 cm$^{-1}$ can be attributed to the typical $(v_2 + v_6) / 2v_6 - OH$ bending feature of $H_2O_2$[42], which gradually increased due to the rapid structural transformation. The $O - O$ characteristic peak at 946 cm$^{-1}$ was clearly detected in the dark reaction, indicating that the protonated anion intermediate promoted oxygen adsorption through the Yeager-type (side-on) absorption (Supplementary Fig. 57). These phenomena confirmed that the anion radical intermediate combined with $O_2$ to produce $H_2O_2$ and then reverted to the carbonyl group, accomplishing a catalytic redox cycle. The entire process is similar to plant photosynthesis involving light and dark reactions.

Isotopic labeling experiments were conducted to investigate photocatalytic $H_2O_2$ production (Fig. 4c and Table S2). With the $H_2^{16}O$ and $^{18}O_2$ system, $^{18}O$ signals were evidently observed in the produced $H_2O_2$, indicating the existence of the $O_2$ reduction pathway for $H_2O_2$ synthesis in our catalytic system[43,44]. The amount of $H_2^{18}O_2$ generated via reduction of $^{18}O_2$ and the total $H_2O_2$ production amount were 0.0652 mmol and 0.1269 mmol, respectively, indicating that half of the $H_2O_2$ was mainly produced by the $O_2$ reduction pathway. Similarly, to detect the O in $H_2O_2$ produced from the $H_2O$ oxidation reaction, we used $H_2^{18}O$ and $^{16}O_2$ as the reaction system. The isotopic results showed that $^{18}O$ signals were also observed in the produced $H_2O_2$, confirming $H_2O_2$ production through $H_2O$ oxidation[43,44]. The amount of $H_2^{18}O_2$ generated via the oxidation of $H_2^{18}O$ and the total $H_2O_2$ production amount were 0.0512 mmol and 0.1040 mmol, respectively, demonstrating that half of the $H_2O_2$ was mainly produced by the $H_2O$ oxidation pathway. Therefore, the amounts of $H_2O_2$ produced by oxidation reaction and reduction reaction are relatively equivalent. Moreover, we also proved that the proton ($H^+$) source for $H_2O_2$ generation was $H_2O$ (Fig. 4d, e). We measured the $^1H$ spectrum of $H_2O_2$ with $D_2O$ as a solvent to qualitatively observe its chemical shift. In $D_2O$, considering the concentration of the products, we prepared a 15 mM $H_2O_2$ solution with $D_2O$ as the solvent. Affected by the solvent, the chemical shift ($^1H$) of $H_2O_2$ in $D_2O$ was 11.04 ppm. These results were consistent with the $^1H$ spectrum of $H_2O_2$ in a previous publication[45]. Then, we carried out the photocatalytic $H_2O_2$ production experiment in $D_2O$. After the reaction, the suspension was monitored with $^1H$ nuclear magnetic resonance (NMR). No $H_2O_2$ signal was observed at 11.04 ppm in the sample, indicating that the proton source for $H_2O_2$ generation was $H_2O$. Elemental analysis was conducted for the PI-BD-TPB photocatalyst before and after the photoreaction, and the H content was basically stable (Supplementary Fig. 58). Thus, photogenerated holes oxidize water to produce $H_2O_2$, and then, $O_2$ combines with $H^+$ induced by the holes and subsequently releases $H_2O_2$ in our system. Nine photocatalysts with carbonyl groups were prepared and evaluated to verify the anion radical intermediate-mediated $H_2O_2$ synthesis strategy (Supplementary Figs. 59–64).

The $O_2$ reduction process over the PI-BD-TPB photocatalyst was simulated to understand the reaction pathways from a thermodynamic point of view. As depicted in Fig. 4f, we compared the Gibbs free energy diagrams of the conventional one-step 2e$^-$ pathway (Pathway I) and the pathway of the photocatalytic redox cycle via the redox intermediate (Pathway II) for $H_2O_2$ production. In Pathway II, the $O_2$* species had a lower Gibbs free energy ($\Delta G = -0.99$ eV) at the two symmetric C atoms of the carbonyl group on the imide ring. The calculation results revealed that the anion radical intermediate-mediated photocatalytic redox cycle was beneficial for $O_2$ adsorption, providing a powerful thermodynamic driving force. Moreover, Pathway II exhibited a lower $\Delta G$ for *HOOH formation ($-2.60$ eV) than Pathway I ($-2.30$ eV) and was more favorable for $H_2O_2$ desorption from the photocatalyst surface. A rotating disk electrode test was performed to determine the average electron transfer number involved in the $O_2$ reduction reaction. The average electron transfer number was calculated to be approximately 1.976, indicating that the PI-BD-TPB aerogel had a high two-electron selectivity for the $O_2$ reduction reaction (Supplementary Fig. 65). To investigate the $H_2O_2$ desorption capability of the PI-BD-TPB aerogel, electrochemical $H_2O_2$ reduction reaction measurements were conducted in an Ar-saturated 0.5 M $H_2SO_4$ electrolyte containing 15 mM $H_2O_2$. We demonstrated that BD-TPB under light irradiation had poorer $H_2O_2$ reduction reaction activity (Supplementary Fig. 66). We also tested the electrochemical performance for $H_2O_2$ generation on PI-BD-TPB photocatalyst via the constant potential method. These results showed that the PI-BD-TPB had excellent two-electron oxygen reduction for $H_2O_2$ electrosynthesis (Supplementary Fig. 67).

Through comprehensive in situ spectroscopic studies correlated with theoretical calculations, we elucidated the reaction pathways over the PI-BD-TPB photocatalyst (Fig. 4g). Under the excitation of simulated sunlight, photoelectrons and holes were effectively separated. Photogenerated holes oxidized water to produce $H_2O_2$ and proton source. Moreover, the $C = O$ group in the imide ring stored photogenerated electrons was converted into the anion radical intermediate, and subsequently combined with $H^+$ produced by holes, transforming into the protonated anion intermediate. The intermediate spontaneously adsorbed $O_2$ to once again release $H_2O_2$ and reverted to the $C = O$ group as in the original state, achieving a catalytic redox cycle.

## Discussion

In summary, a crosslinked polyimide aerogel photocatalyst incorporating reductive $C = O$ groups was engineered for the effective synthesis of $H_2O_2$ from $H_2O$ and $O_2$. With an AQY of 14.28% at $420 \pm 10$ nm, the polyimide aerogel under 100 mW cm$^{-2}$ irradiation for 8 h produced 88.6 mmol m$^{-2}$ $H_2O_2$. More importantly, our research introduces the approach for $H_2O_2$ photosynthesis involving an aromatic anion radical intermediate that mediates the photocatalytic redox cycle with $O_2$. This redox cycle pathway not only dramatically improves $O_2$ adsorption but also thermodynamically favors the oxygen reduction reaction, consequently enhancing the efficiency of $H_2O_2$ photosynthesis.

## Methods

### Synthesis of polyimide BD-TPB aerogel

All chemicals used in the research were purchased from Tansoole Co, Ltd., without additional purification. The polyimide BD-TPB photocatalyst presented in this research were prepared by the imidization of aromatic triamines 1,3,5-tris[4-amino(1,1-biphenyl-4-yl)] benzene (TPB) with aromatic dianhydrides 3,3',4,4'-biphenyltetracarboxylic dianhydride (BD). Typically, BD (26.47 mg, 0.90 mmol) dissolved in 1.0 mL of 1-methayl-2-pyrrolidinone (NMP) solution, then TPB (34.78 mg, 0.60 mmol) dispersed in 1.0 mL of Mesitylene solution, then Isoquinoline (0.1 mL) was added. After being sonication with a power of 100 W for 10 min, and luminous yellow solution was quickly obtained. Then poured into a mold and allowed to gel within 60 min in 0 °C. The gel was upon further solvothermal treatment at 180 °C for 48 h. The products were washed in a solution of 75% NMP in ethanol for 24 hours. Subsequently, the solvent was exchanged in 24 h intervals with 25% NMP in ethanol, and then 100% ethanol for three times. Finally, the macroscopic BD-TPB aerogel was rinsed with deionized water and was freeze-dried (yield: ~92%).

## Synthesis of poly (amic acid) (PAA)

The precursor of polyimide was synthesized from the polycondensation reaction between the imidization of the dianhydrides BD and the triamines TPB. BD (26.47 mg, 0.90 mmol) was dissolved in 2.0 mL NMP solution, then TPB (34.78 mg, 0.60 mmol) was also added. The mixture solution was stirred at 0 °C for 24 h to obtain the yellow solution named PAA solution. Then PAA solution was precipitated with deionized water and the precipitate was washed several times with ethanol and dried under vacuum (yield: ~ 65%).

## Scalable synthesis of polyimide BD-TPB aerogel membrane

Typically, BD (2.64 g) dissolved in 200 mL NMP solution, then TPB (3.47 g) dispersed in 200 mL Mesitylene solution, then Isoquinoline (1.0 mL) was added. After being sonication with a power of 100 W for 10 min, and luminous yellow solution was quickly obtained. Then poured into the mold and allowed to gel within 12 h in 0 °C. The gel was upon further solvothermal treatment at 180 °C for 48 h. The products were washed in a solution of 75% NMP in ethanol for 24 hours. Subsequently, the solvent was exchanged in 24 h intervals with 25% NMP in ethanol, and then 100% ethanol for three times. Finally, the macroscopic BD-TPB aerogel membrane was dried (yield: ~88%).

## Photocatalytic $H_2O_2$ production

15 mg of PI-BD-TPB photocatalyst and 20 mL of deionized water were put in a quartz square bottle (100 mL). $O_2$ was bubbled into the suspension for 30 min in the dark. During the reaction, $O_2$ was kept bubbling to maintain the $O_2$-rich environment. A 300 W Xe lamp was utilized as the light source, and all photocatalytic experiments were performed in the same experimental condition. The light average intensity is 325 mW·cm$^{-2}$. The $H_2O_2$ concentration was determined by a potassium titanium oxalate method[46].

## Determination of AQY efficiency

The apparent quantum yield (AQY) of photocatalyst was measured under 300 W Xe lamp irradiation. The photocatalytic reaction was carried out with 40 mg photocatalyst powder in 25 mL water at 50 °C. The active area of the reactor was ~ 0.785 cm$^{-2}$. The light intensity at 420 nm ± 10 nm was calculated to be 3.72 mW cm$^{-2}$. Then, AQY was calculated by the following equation:

$$AQY(\%) = (N_{H_2O_2 formed(mol)})/(N_{photons}) \times 100\% \qquad (1)$$

## Measurement of SCC efficiency

The solar-to-chemical energy conversion (SCC) efficiency was determined by the photocatalytic experiment[47]. 80 mg photocatalyst powder and 60 mL water were added into a flask and bubbled with $O_2$ for 30 min, the reaction was carried out at 50 °C in the water bath. The SCC efficiency was calculated via the following equation: where the free energy (ΔG) for $H_2O_2$ formation is 117 kJ mol$^{-1}$, the irradiance of the spectrum is 1000 W m$^{-2}$ and the irradiated area is $0.785 \times 10^{-4}$ m$^2$. The total input energy was therefore 0.0785 W. reaction time is 1200 s. The SCC was calculated as follows:

$$SCC(\%) = (\Delta G \times H_2O_2 formed(mol))/(I \times A \times t) \times 100\% \qquad (2)$$

## The stability test

The PI-BD-TPB photocatalyst and 20 mL of deionized water were used in a quartz square bottle. The suspension was well dispersed by ultrasonication for 15 min and $O_2$ was bubbled into the suspension for 30 min in the dark. During the reaction, oxygen was kept bubbling to maintain the $O_2$-rich environment. Other conditions were kept the same. Between each test, $H_2O_2$ and $H_2O$ were removed by evaporation.

After that, the left macroscopic PI-BD-TPB photocatalyst was further dried in a vacuum dry box at 60 °C for 12 h to remove the possible residual of $H_2O_2$.

## Photocurrents and photoelectrochemical measurements

The Mott–Schottky plots and electrochemical impedance of the catalysts were measured on an electrochemical workstation (CHI660E, CHI Instruments, Shanghai, China)[48]. A 300 W Xe lamp was utilized as the light source and $Na_2SO_4$ (0.5 M) aqueous solution was used as the supporting electrolyte[49]. A platinum wire and Ag/AgCl electrode were used as counter electrode and reference electrode. 50 μL of Nafion, dry ethanol (1 mL) and photocatalyst (5 mg) were sonicated for 30 min. Then 100 μL of the suspension was dripped onto an ITO glass substrate and dried. The application potential was converted to RHE potentials with respect to Ag/AgCl using the following equation:

$$E_{(vs. RHE)} = E_{(vs. Ag/AgCl)} + 0.197V + 0.0591*pH \qquad (3)$$

## Rotating disk electrode measurement

A glassy carbon rotating disk electrode was served as the substrate for the working electrode. The working electrode was prepared as follows: 20 mg of power photocatalysts was dispersed in 2 mL ethanol containing 20 μL Nafion. 20 μL of the above slurry was put onto the disk electrode and dried at room temperature. The linear sweep voltammogram curves were recorded in an $O_2$-saturated 0.1 M phosphate buffer solution at room temperature and a scan rate of 10 mV s$^{-1}$ with different rotation speeds.

## Rotating ring-disk electrode measurement

A ring-disk electrode was served as the substrate for the working electrode. The voltammograms were obtained in a 0.1 M phosphate buffer solution under Ar atmosphere at a scan rate of 10 mV s$^{-1}$ and a rotation rate of 1000 rpm. The potential of the ring electrode was set to − 0.23 V (vs. Ag/AgCl) to detect $O_2$. The potential of the ring electrode was set to 0.6 V (vs. Ag/AgCl) to detect $H_2O_2$.

## Electron paramagnetic resonance measurement

The EPR measurement was carried out to detect superoxide radical or hydroxide radical by adding 5,5-dimethyl-1-pyrroline N-oxide (DMPO) as a spin-trapping reagent. A 300 W Xe lamp was used as the light source. The measurements were conducted as follows: photocatalyst were dispersed in water or MeOH containing DMPO with a Pyrex glass tube which was sealed with a rubber septum cap.

## Isotopic labeling experiment

$^{18}O$ in the produced $H_2^{18}O_2$ was determined by converting $H_2^{18}O_2$ to $H_2^{18}O$[46].$^{18}O$ in the converted $H_2O$ was analyzed by Liquid Water Isotope Analyzer (Los Gatos Research, USA). The conversion method was as follows: First, the collected potassium titanium oxalate-$H_2O_2$ complex was re-dissolved and dried; then, the potassium titanium oxalate-$H_2O_2$ complex was reduced by KI to convert the contained $H_2O_2$ species to $H_2O$; finally, the converted $H_2O$ was collected by distillation and detected by Liquid Water Isotope Analyzer.

## Data availability

The authors declare that all data supporting the findings of this study are available within the paper, Supplementary Information files and source data at the figshare link.

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

## Acknowledgements

The work is supported by the National Natural Science Foundation of China (22136002, 22172064, 22376083), Special Fund Project of Jiangsu Province for Scientific and Technological Innovation in Carbon Peaking and Carbon Neutrality (BK20220023). The authors would also like to thank Yaning Zhang, Yong Liu, Yunfan Yang, and Yujie Ling from Jiangnan University for their help.

## Author contributions

W.C. and Y.D. contributed equally. Y.D. and Y.Z. conceived the idea. W.C. conducted performance evaluations, designed experiments, and fabrication experiments. B.L. and Z. L. conducted the theoretical calculation. C.P., J.Z., and H.Z. participated in the paper discussions. W.C., Y.D. and Y.Z. planned the research and wrote the paper. All the authors revised and approved the manuscript.

## Competing interests

The authors declare no competing interests.
