## [Peer Review File · Nature Communications]

A photocatalytic redox cycle over a polyimide catalyst drives efficient solar-to-H₂O₂ conversionREVIEWER COMMENTS

Reviewer #1 (Remarks to the Author):

The manuscript presents an innovative approach to solar-driven photosynthesis of hydrogen peroxide (H₂O₂) from water and oxygen, utilizing a polyimide aerogel photocatalyst with photo-reductive carbonyl groups. The authors propose a novel mechanism wherein the carbonyl groups on the photocatalyst surface are reduced under photoexcitation, leading to the generation of an anionic radical intermediate. This intermediate undergoes photocatalytic redox cycling with oxygen to accelerate H₂O₂ synthesis. The manuscript highlights significant performance improvements, including a high H₂O₂ generation rate (2.85 mM h⁻¹), exceptional cycle stability (above 144 hours), and notable solar-to-chemical conversion efficiency (0.92%). Moreover, the photocatalyst demonstrates promising application feasibility, achieving a high photocatalytic yield (34.3 mmol m⁻²) under natural sunlight. Various spectroscopic techniques and DFT calculations provide experimental evidence supporting the proposed mechanism. The manuscript presents a significant contribution to the field of solar-driven H₂O₂ photosynthesis, offering both innovative insights and practical implications for sustainable technologies. The comprehensive experimental evidence, coupled with the impressive performance metrics, strengthens the manuscript's credibility. Therefore, I recommend acceptance of this manuscript for publication in Nature Communications, contingent upon addressing any minor issues or clarifications identified during the review process.

1. The consideration of the carbonyl group as an oxygen reduction site in this manuscript represents a departure from previous reports where it was identified as an excellent water oxidation site (Angew. Chem. Int. Ed. 2014, 53, 1 - 7; ACS Catal. 2016, 6, 7021-7029.). While previous studies have indeed highlighted the potential of carbonyl groups in imides for water oxidation, the current manuscript proposes a novel mechanism wherein the carbonyl groups on the surface of the polyimide aerogel photocatalyst are reduced under photoexcitation to generate an anionic radical intermediate. This intermediate then participates in a photocatalytic redox cycling process with oxygen, ultimately leading to the synthesis of hydrogen peroxide (H₂O₂). The authors have likely provided experimental evidence and theoretical considerations to support their proposition that the carbonyl groups act as sites for oxygen reduction in the context of their photocatalytic system. More detailed discussions are required here.

2. Including solid-state ¹³C NMR spectra after long-term recycling of the catalyst would be beneficial in demonstrating the recovery of the carbonyl functional group within the structure. This supplementary data would provide valuable insights into the stability and integrity of the catalyst's structure over extended periods of use, further supporting the conclusions drawn in the manuscript. Therefore, I recommend that the authors consider including this additional experimental data to enhance the comprehensiveness of their study.

3. While the manuscript demonstrates the two-electron water oxidation reaction occurring at the hole terminal in detail, the observed decrease in the concentration of hydrogen peroxide in Figure S10 raises a pertinent question. Further investigation into the factors influencing hydrogen peroxide concentration is warranted to address this discrepancy. Potential explanations may include side reactions, catalyst deactivation, or limitations in the reaction kinetics. Therefore, the authors should provide a thorough

analysis or discussion addressing the observed decrease in hydrogen peroxide concentration, ensuring the clarity and coherence of their findings.

4. For convenient comparison with other reported units of photocatalytic hydrogen peroxide production, the manuscript should include rate curves of the PI-BD-TPB photocatalysts under simulated sunlight and visible light irradiation for one hour. This additional data would allow readers to assess the performance of the photocatalysts under different light conditions and facilitate meaningful comparisons with existing literature. Therefore, I recommend that the authors include these rate curves in their manuscript to enhance its completeness and relevance to the broader scientific community.

5. The observation that the impedance in oxygen atmosphere was lower than in argon atmosphere, as indicated in Figure S41, suggests potential differences in the photocurrent behavior under different atmospheres. To ensure the accuracy and reliability of the results, it would be valuable to conduct photocurrent tests in both oxygen and argon atmospheres. This additional experimental data would provide insights into the influence of atmospheric conditions on the photocurrent response of the photocatalyst, thereby enhancing the robustness of the study's conclusions. Therefore, I recommend that the authors include photocurrent tests in different atmospheres to complement their impedance measurements and strengthen the validity of their findings.

6. To determine the voltage value of the electron transfer number during oxygen reduction reactions, Koutecky-Levich curves should be provided. These curves would allow for the quantitative determination of the electron transfer number and provide valuable insights into the kinetics of the oxygen reduction reaction on the photocatalyst surface. Therefore, I recommend that the authors include Koutecky-Levich curves in their manuscript to further elucidate the mechanism of oxygen reduction and enhance the rigor of their study.

Reviewer #2 (Remarks to the Author):

In this work, the authors designed and synthesized a polyimide aerogel photocatalyst with photo-reductive carbonyl group, achieving an innovative oxygen reduction pathway for H₂O₂ photosynthesis. This pioneering route involves a photocatalytic redox cycling mechanism orchestrated by anionic radical intermediates, culminating in a remarkable apparent quantum yield (AQY) of 14.28% and the solar-to-chemical conversion (SCC) efficiency of 0.92%. Overall, the manuscript provides a greatly detailed analysis of the mechanism of anion radical intermediate-mediated photocatalysis redox cycle for H₂O₂ production, through a combination of in-situ characterization and DFT calculations, which is extremely interesting and convincing. However, some minor revision is necessary for publication in Nat. Commun. The detailed comments are as follows:

1. The organic photocatalysts are extremely susceptible to autoxidation, and this work qualitatively showed that the two-electron water oxidation reaction occurs at the hole. However, whether the pathway of four-electron water oxidation to produce oxygen would also co-exist for the holes in this manuscript, which required additional demonstration.
2. The light-dark experiment of Figuer4a was performed under pure water conditions, with weak phenomena of dark reaction. When performed with sacrificial reagent such as methanol or benzyl alcohol, would the production of hydrogen peroxide from the dark reaction be enhanced via the

availability of sufficient photogenerated electrons and protons?

3. In spite of this research focusing on the novel mechanism of H₂O₂ photosynthesis, the separation of charges during photocatalysis is an essential factor. Therefore, the related photophysical characterization, such as surface photovoltage experiment and transient fluorescence spectroscopy should be performed.
4. The AQY should be complemented in the range of 500-700 nm, such as providing 530 nm, 600 nm and 650 nm quantum efficiency, and the accuracy of the data should be reflected by the error bars.
5. The stability of photocatalysts is an important reference for practical applications, where the morphology and structure after photocatalysis is insufficient to prove the stability of the material in this manuscript, which should be added with ultraviolet absorption spectra after long-term reaction to prove the stable photophysical properties.
6. The specific light intensity of the source employed in all photocatalytic experiments in the manuscript should be provided.

Reviewer #3 (Remarks to the Author):

In this submission, the authors synthesized a polyimide aerogel photocatalyst with carbonyl group (PI-BD-TPB) to generate hydrogen peroxide (H₂O₂). The photoreduction of the carbonyl group into anion radical intermediate promotes the adsorption of O₂, thereby boosting the catalytic performance. While the concept of this work is interesting, reasonable explanations of the reaction mechanism and experimental results are lacking. More data and relative discussion should be added to make the conclusion more convincing.

1. This article reported that both electron reduction and hole oxidation produce H₂O₂. What is the stoichiometric ratio of H₂O₂ produced by oxidation and reduction?
2. The EPR signal in Fig. 2f was described as imide radical. Why? This EPR signal may also correspond to defects in the catalyst.
3. The authors said that photogenerated-hole can oxidize water to produce H₂O₂. What active species are involved during this process? Please provide experimental evidence.
4. Which specific anion radical does the signal at 418 nm correspond to (Fig. 2g)? Further experimental evidence and detailed structural formulas should be given. In addition, the role of Na₂S₂O₄ also needs to be explained in detail.
5. Supplementary Fig.15-36 should be added to the manuscript with reasonable statements and explanations.
6. Is this anion radical intermediate-mediated H₂O₂ synthesis strategy universal? Please test at least five other photocatalysts with carbonyl groups for evaluation.
7. Under actual electrochemical reaction conditions, the surface charge of the catalyst varies with the electrode potential. Therefore, a more appropriate calculation method (constant potential approach) and solvation models should be used to describe the real electrochemical behavior during the synthesis of H₂O₂.

Response to Reviewers

Manuscript Title: H₂O₂ Synthesis with Solar-to-Chemical Conversion 0.92% via Anion Radical Intermediate by Photocatalytic Redox Cycle of Polyimide

Manuscript Number: NCOMMS-24-03300A

Author: Wenwen Chi⁺, Yuming Dong^{+*} and Yongfa Zhu^{*}

[The reviewer comments are shown in *italic*]

[Responses with the emphasized parts are in **Blue**]

[Revisions in the manuscript and supplementary information are highlighted in **blank**]

Reviewer #1 (Remarks to the Author):

The manuscript presents an innovative approach to solar-driven photosynthesis of hydrogen peroxide (H_2O_2) from water and oxygen, utilizing a polyimide aerogel photocatalyst with photo-reductive carbonyl groups. The authors propose a novel mechanism wherein the carbonyl groups on the photocatalyst surface are reduced under photoexcitation, leading to the generation of an anionic radical intermediate. This intermediate undergoes photocatalytic redox cycling with oxygen to accelerate H_2O_2 synthesis. The manuscript highlights significant performance improvements, including a high H_2O_2 generation rate (2.85 mM h^{-1}), exceptional cycle stability (above 144 hours), and notable solar-to-chemical conversion efficiency (0.92%). Moreover, the photocatalyst demonstrates promising application feasibility, achieving a high photocatalytic yield (34.3 mmol m^{-2}) under natural sunlight. Various spectroscopic techniques and DFT calculations provide experimental evidence supporting the proposed mechanism. The manuscript presents a significant contribution to the field of solar-driven H_2O_2 photosynthesis, offering both innovative insights and practical implications for sustainable technologies. The comprehensive experimental evidence, coupled with the impressive performance metrics, strengthens the manuscript's credibility. Therefore, I recommend acceptance of this manuscript for publication in *Nature Communications*, contingent upon addressing any minor issues or clarifications identified during the review process.

Response: We appreciate you very much for the valuable comments.

1.Comment: The consideration of the carbonyl group as an oxygen reduction site in this manuscript represents a departure from previous reports where it was identified as an excellent water oxidation site (*Angew. Chem. Int. Ed.* 2014, 53, 1-7; *ACS Catal.* 2016, 6, 7021-7029.). While previous studies have indeed highlighted the potential of carbonyl groups in imides for water oxidation, the current manuscript proposes a novel mechanism wherein the carbonyl groups on the surface of the polyimide aerogel photocatalyst are reduced under photoexcitation to generate an anionic radical intermediate. This intermediate then participates in a photocatalytic redox cycling process with oxygen, ultimately leading to the synthesis of hydrogen peroxide (H_2O_2). The authors have likely provided experimental evidence and theoretical considerations to support their proposition that the carbonyl groups act as sites for oxygen reduction in the context of their photocatalytic system. More detailed discussions are required here.

Response: Thank you for your thoughtful advice, we would like to explain above-mentioned issue in detail as follows:

- a) As a type of known polymers, polyimide (PI) materials contain enriched carbonyl functional groups (*Angew. Chem.* 2020, 59, 18322–18333; *Angew. Chem.* 2020, 132, 18478–18489). Due to the strong electronegativity of oxygen atom, carbonyl groups in polyimide have been reported to act as active sites. Meanwhile, carbonyl groups from polyimide have the property of absorbing and storing electrons, which not only promotes the directional migration of electrons but also reduces the

recombination rate of carriers. Moreover, carbonyl groups in imide ring are connected to the donor-electron unit via the conjugated imide bond, which is beneficial for charge transport (*Small* **2021**, 17, 2005752; *Angew. Chem.* **2018**, 57, 9443; *Chem. Rev.* **2016**, 116, 11685.). Therefore, carbonyl group of polyimide is a promising reduction site.

- b) The results of theoretical calculation showed that the highest occupied molecular orbital was mainly distributed on the TPB unit, while the lowest unoccupied molecular orbital was mainly distributed on the BD unit. This implies that the BD unit has electron-accepting property and further forms the donor-acceptor structure in the PI-BD-TPB aerogel (**Supplementary Figure S38**). Carbonyl groups of PI-BD-TPB photocatalyst have robust electron affinity characteristic in the **Supplementary Figure S39** of electrostatic potential distribution and in the **Supplementary Figure S40** of electron distributions under photoexcitation, showing that carbonyl groups tend to accept photogenerated electrons. Thus, carbonyl groups of PI-BD-TPB are served as reductive center.
- c) We performed cyclic voltammetry measurements to define reduction potential of carbonyl groups. As shown in **Supplementary Figure S41**, two pairs peak at approximately 0.509 V / 1.674 V and 0.009 V / 0.905 V (vs. RHE) are clearly observed. The above phenomenon can be attributed to enolization reaction of carbonyl oxygen, indicating that carbonyl groups in imide ring storage electron to generate anion radical. PI-BD-TPB ($E_{CB} = -0.19$ V vs. RHE) thermodynamically supports the reduction of carbonyl groups to anion radical (0.509, 0.009 V vs RHE). Therefore, the reduction of carbonyl groups to anion radical is extremely likely to occur in our photocatalytic process. Combined *in situ* X-ray photoelectron spectroscopy spectrum of O 1s (**Figure 2e**), electron paramagnetic resonance spectra (**Figure 2f**), ultraviolet-visible absorption spectra (**Figure 2g**), fluorescence test (**Supplementary Figure S49**) with related literatures, we also confirmed the formation of the imide anion radical.
- d) *In situ* FTIR spectra, *in situ* Raman spectra and further theoretical calculations were performed to explore of the role of anion intermediates in oxygen reduction reaction (**Figure 3a-3e** and **Figure 4f**). Through comprehensive calculations and a series of *in situ* techniques, we have demonstrated that anion intermediates effectively promote oxygen adsorption, and thermodynamically reduce the energy barrier for two-electron oxygen reduction reaction.

Therefore, we confirmed this process that carbonyl groups on the polyimide aerogel

photocatalyst surface were reduced to convert into the anion radical intermediate under photoexcitation. The intermediate spontaneously adsorbed O₂ to produce H₂O₂ generation, and reverted to the carbonyl groups, accomplishing a catalytic redox cycle.

2.Comment: Including solid-state ¹³C NMR spectra after long-term recycling of the catalyst would be beneficial in demonstrating the recovery of the carbonyl functional group within the structure. This supplementary data would provide valuable insights into the stability and integrity of the catalyst's structure over extended periods of use, further supporting the conclusions drawn in the manuscript. Therefore, I recommend that the authors consider including this additional experimental data to enhance the comprehensiveness of their study.

Response: According to your professional suggestion, we have added the solid-state ¹³C NMR spectroscopy after the photocatalytic reaction. Related results and discussions were supplemented in the revised Supplementary Information (**Supplementary Figure S8**). The peak at 166.5 ppm for the fresh PI-BD-TPD photocatalyst and 166.4 ppm for the used photocatalyst were assigned to the carbon of carbonyl group, indicating the recovery of carbonyl group. Other overlapping peaks in the range 100-150 ppm for the fresh and used PI-BD-TPD were ascribed to the carbon of benzene ring. Therefore, the chemical structure and carbonyl groups of PI-BD-TPB exhibited negligible changes before and after photoreaction, showing that PI-BD-TPB had the excellent photostability.

The solid-state ¹³C NMR spectra of the PI-BD-TPB photocatalyst after long-term reaction.

3.Comment: *While the manuscript demonstrates the two-electron water oxidation reaction occurring at the hole terminal in detail, the observed decrease in the concentration of hydrogen peroxide in **Figure S10** raises a pertinent question. Further investigation into the factors influencing hydrogen peroxide concentration is warranted to address this discrepancy. Potential explanations may include side reactions, catalyst deactivation, or limitations in the reaction kinetics. Therefore, the authors should provide a thorough analysis or discussion addressing the observed decrease in hydrogen peroxide concentration, ensuring the clarity and coherence of their findings.*

Response: According to your meticulous suggestion, we would like to explain this issue from three areas:

(a) The PI-BD-TPB aerogel was no significant structural and morphological changes after the long-term photocatalytic reaction according to FTIR, solid-state ¹³C NMR, XRD, SEM and TEM results (**Supplementary Figure S7-10**). These results show that PI-BD-TPB aerogel has no problem of deactivation during the photoreaction and has good photostability.

(b) The PI-BD-TPB photocatalyst underwent charge separation under argon atmosphere and light irradiation, and photogenerated hole oxidized water to produce hydrogen peroxide. Because photogenerated electrons cannot be consumed in time, hydrogen peroxide produced by photogenerated hole is a small amount. Meanwhile, photogenerated electrons may decompose the additional hydrogen peroxide, causing decrease in the concentration of hydrogen peroxide.

(c) According to relevant literature (*Angew. Chem. Inter. Ed.* **2023**, 62, e202309624; *Applied Catalysis B: Environmental* **2022**, 316, 121675), UV light easily decomposes hydrogen peroxide. The light source by we used was simulated sunlight containing UV light in decomposition experiment, which also caused a decrease in the concentration of H₂O₂.

Therefore, combined with the above analysis, the concentration of H₂O₂ slightly decrease in our system.

4.Comment: *For convenient comparison with other reported units of photocatalytic hydrogen peroxide production, the manuscript should include rate curves of the PI-BD-TPB photocatalysts under simulated sunlight and visible light irradiation for one hour. This additional data would allow readers to assess the performance of the photocatalysts under different light conditions and facilitate meaningful comparisons with existing literature. Therefore, I recommend that the authors include these rate*

curves in their manuscript to enhance its completeness and relevance to the broader scientific community.

Response: According to your detailed advice, we have supplemented the rate curves of PI-BD-TPB under simulated sunlight and visible light irradiation in O₂ atmosphere. Activity experiments were performed in 15 mg photocatalyst and 20 mL water system. Reaction solutions were taken at 20 min intervals to test the concentration of H₂O₂. As presented in the following figure, the PI-BD-TPB photocatalyst presented H₂O₂ production rate of 3777.33 $\mu\text{mol g}^{-1} \text{h}^{-1}$ under simulated sunlight and 2632.88 $\mu\text{mol g}^{-1} \text{h}^{-1}$ under visible light irradiation, respectively (**Supplementary Figure S4**). Therefore, PI-BD-TPB still has a high H₂O₂ generation efficiency under visible light irradiation.

Time-dependent H₂O₂ rate curves of the PI-BD-TPB under different light sources.

5.Comment: The observation that the impedance in oxygen atmosphere was lower than in argon atmosphere, as indicated in **Figure S41**, suggests potential differences in the photocurrent behavior under different atmospheres. To ensure the accuracy and reliability of the results, it would be valuable to conduct photocurrent tests in both oxygen and argon atmospheres. This additional experimental data would provide insights into the influence of atmospheric conditions on the photocurrent response of the photocatalyst, thereby enhancing the robustness of the study's conclusions. Therefore, I recommend that the authors include photocurrent tests in different atmospheres to complement their impedance measurements and strengthen the validity of their findings.

Response: Following your professional guidance, we have added photocurrent spectra of the PI-BD-TPB under different atmosphere and further understood the influence of atmosphere condition on the photocurrent response of photocatalyst. The conditions of photocurrent test were consistent with electrochemical impedance spectroscopic spectra. A 300 W Xe lamp was utilized as the light source and Na₂SO₄ (0.5 M) aqueous solution was used as electrolyte. A platinum wire and Ag/AgCl electrode were used as counter electrode and reference electrode. The photocurrent of PI-BD-TPB in oxygen atmosphere was approximately four times higher than that in argon atmosphere, indicating that photogenerated carriers effectively separated in oxygen atmosphere (**Supplementary Figure S45**). That was consistent with the result of having smaller impedance for PI-BD-TPB in oxygen atmosphere. The above phenomenon is attributed to the fact that anion intermediates form in light irradiation and further undergo reduction reaction with oxygen, which efficiently promotes charge separation.

Photocurrent spectra of the PI-BD-TPB under Ar or O_2 atmosphere.

6.Comment: To determine the voltage value of the electron transfer number during oxygen reduction reactions, Koutecky-Levich curves should be provided. These curves would allow for the quantitative determination of the electron transfer number and provide valuable insights into the kinetics of the oxygen reduction reaction on the photocatalyst surface. Therefore, I recommend that the authors include Koutecky-Levich curves in their manuscript to further elucidate the mechanism of oxygen

reduction and enhance the rigor of their study.

Response: Regarding the question you raised, we would like to explain this issue in detail. We performed rotating disk electrode test to determine the average electron transfer number involved in the O₂ reduction reaction. The average electron transfer number was calculated to be approximately 1.976 and the corresponding potential at -0.6 V (vs. Ag / AgCl). For this part of the content improvement, we have added in the supporting information (**Supplementary Figure S64**). We also added relevant content to the revised manuscript on page 15: A rotating disk electrode test was performed to determine the average electron transfer number involved in the O₂ reduction reaction. The average electron transfer number was calculated to be approximately 1.976, indicating that the PI-BD-TPB aerogel had a high two-electron selectivity of the O₂ reduction reaction.

Reviewer #2 (Remarks to the Author):

In this work, the authors designed and synthesized a polyimide aerogel photocatalyst with photo-reductive carbonyl group, achieving an innovative oxygen reduction pathway for H₂O₂ photosynthesis. This pioneering route involves a photocatalytic redox cycling mechanism orchestrated by anionic radical intermediates, culminating in a remarkable apparent quantum yield (AQY) of 14.28% and the solar-to-chemical conversion (SCC) efficiency of 0.92%. Overall, the manuscript provides a greatly detailed analysis of the mechanism of anion radical intermediate-mediated photocatalysis redox cycle for H₂O₂ production, through a combination of in-situ characterization and DFT calculations, which is extremely interesting and convincing. However, some minor revision is necessary for publication in Nat. Commun. The detailed comments are as follows:

Response: We appreciate you very much for your kind comments.

1. Comment: The organic photocatalysts are extremely susceptible to autoxidation, and this work qualitatively showed that the two-electron water oxidation reaction occurs at the hole. However, whether the pathway of four-electron water oxidation to produce oxygen would also co-exist for the holes in this manuscript, which required additional demonstration.

Response: On the basis of your profound insights, we would like to comprehensively explain above-mentioned issue from three areas and provide a series of experimental evidence.

(a) We performed H₂¹⁸O isotopic labeling experiments to investigate the photocatalytic H₂O₂ production via the water oxidation half-reaction (**Supplementary Scheme S1** and **Table S2**). Isotopic experiment results showed that the amount of H₂¹⁸O₂ in the H₂¹⁸O and ¹⁶O₂ system was almost 50% of hydrogen peroxide production in total, confirming that the two-electron water oxidation process occurs in our photocatalyst system.

(b) We checked the selectivity of water oxidation process by adding La (NO₃)₃ as electron scavenger to inhibit the reduction half-reaction. H₂O₂ generation experiments were performed in Ar-saturated water with 2 mM La (NO₃)₃; O₂ evolution experiments were performed in Ar-saturated water with 2 mM La (NO₃)₃ and 50 mg La₂O₃ as pH buffer. Both experiments were performed in a sealed reactor for 4 h irradiation (Photocatalyst: 30 mg, Solution volume: 20 mL, Temperature: 298 K, Light source: 300 W Xe lamp with as the simulated sunlight). According to these results, negligible O₂ was detected, while 0.388 mM H₂O₂ produced. It proves that photogenerated holes tend to oxidize water to produce H₂O₂ rather than O₂ (**Supplementary Figure S51**).

(c) The PI-BD-TPB aerogel was no significant structural and morphological changes after long-term photoreaction according to the result of FTIR, solid-state ^{13}C NMR, XRD, SEM and TEM (**Supplementary Figure S7-10**). This indicates that the PI-BD-TPB aerogel has no problem of deactivation during the reaction. Therefore, we confirm that photogenerated holes oxidize water to produce H_2O_2 in our system, rather than O_2 evolution.

H_2O_2 production or O_2 evolution in Ar-saturated water with 2 mM $\text{La}(\text{NO}_3)_3$.

2. Comment: The light-dark experiment of **Figuer 4a** was performed under pure water conditions, with weak phenomena of dark reaction. When performed with sacrificial reagent such as methanol or benzyl alcohol, would the production of hydrogen peroxide from the dark reaction be enhanced via the availability of sufficient photogenerated electrons and protons?

Response: According to the professional comments, we have added the “light-dark” experiment by using different hole-sacrificial reagent (Ar-saturated water; Ar-saturated methanol (MeOH) solution 10%; Ar-saturated benzyl alcohol solution 10%, Ar-saturated 5 mM ascorbic acid (Vc) solution). Both experiments were performed in a sealed reactor for 60 min irradiation (Photocatalyst: 30 mg, Solution volume: 20 mL, Temperature: 298 K, Light source: 300 W Xe lamp with as the simulated sunlight). With light off, we immediately purged O_2 gas into reaction system for 30 min. Benzyl alcohol as hole-sacrificing reagent mixed with potassium titanium oxalate after photoreaction, and produced absorption at 400nm, which seriously affected the accuracy of the experimental results. Thus, related experiment results were not discussed. We pay more attention to the results of methanol or ascorbic acid as hole-

sacrificing reagents. Compared to water environment, the concentration of H₂O₂ produced in the dark reaction was increased to 1.82 times with methanol as hole-sacrificial reagent, and 5.69 times with ascorbic acid as the hole-sacrificial reagent. Due to more electrons and protons provided, the amount of H₂O₂ production in dark reaction was boosted by adding hole-sacrificing reagent.

H₂O₂ production in dark reaction by using different hole-sacrificing reagent.

3. Comment: In spite of this research focusing on the novel mechanism of H₂O₂ photosynthesis, the separation of charges during photocatalysis is an essential factor. Therefore, the related photophysical characterization, such as surface photovoltage experiment and transient fluorescence spectroscopy should be performed.

Response: According to your professional suggestion, we have added the surface photovoltage spectroscopy and transient fluorescence spectroscopy in the revised supporting information. The photovoltage intensity of PI-BD-TPB was 35.2 µV, showing that PI-BD-TPB effectively enhanced charge separation (**Supplementary Figure S46**). What's more, the average relaxation lifetime of PI-BD-TPB was 4.08 ns, displaying that PI-BD-TPB had the prolonged lifetime of charge carrier (**Supplementary Figure S47**). Combined with a series of photophysical characterization, we confirm that PI-BD-TPB aerogel with donor-acceptor structure has excellent charge separation efficiency and extended carrier lifetime.

Surface photovoltage spectra of the PI-BD-TPB photocatalyst.

Transient fluorescence spectra of the PI-BD-TPB photocatalyst.

4. Comment: The AQY should be complemented in the range of 500-700 nm, such as providing 530 nm, 600 nm and 650 nm quantum efficiency, and the accuracy of the data should be reflected by the error bars.

Response: According to your detailed comments, we further supplemented the related data into the revised manuscript. We have provided wavelength-dependent AQY spectra represented by error bars for PI-BD-TPB photocatalyst. The AQY experiment of PI-BD-TPB was performed via different incident light wavelengths irradiation, acquiring 13.56%, 14.28%, 7.28%, 3.52%, 2.98%, 1.92% and 1.25% at 400, 420, 435, 500, 530, 600 and 650 nm. The AQY results well matched with the trend of the absorption spectrum, implying the importance of light absorption in the photoreaction (**Fig. 1d**).

Wavelength dependent AQY of PI-BD-TPB in the photocatalytic H₂O₂.

5. Comment: The stability of photocatalysts is an important reference for practical applications, where the morphology and structure after photocatalysis is insufficient to prove the stability of the material in this manuscript, which should be added with ultraviolet.

Response: According to your kind advice, we have supplemented UV-vis diffuse reflectance spectrum after long-term reaction to estimate the photostability for the PI-BD-TPB. No obvious decline or shift in absorption edge was observed after long-term reaction, which demonstrated excellent photocatalytic stability of PI-BD-TPB. Combined with FTIR (**Supplementary Figure S7**), solid-state ¹³C NMR

(Supplementary Figure S8), XRD (Supplementary Figure S9), SEM, TEM (Supplementary Figure S10) and UV-vis absorption results (Supplementary Figure S11), PI-BD-TPB aerogel was no significant structural, morphological and absorption edge changes after photocatalytic reaction. These results shows that PI-BD-TPB aerogel has excellent photostability.

UV-vis diffuse reflectance spectrum PI-BD-TPB aerogel after long-term reaction.

6. Comment: *The specific light intensity of the source employed in all photocatalytic experiments in the manuscript should be provided.*

Response: According to detailed comment, we have added the related experimental detail in the revised supporting information. Corresponding changes were made in the “General methods” section of the revised supporting information as follows: A 300 W Xe lamp was utilized as the light source, and all photocatalytic experiments were performed in the same conditions. The light average intensity is $325 \text{ mW}\cdot\text{cm}^{-2}$.

Reviewer #3 (Remarks to the Author):

In this submission, the authors synthesized a polyimide aerogel photocatalyst with carbonyl group (PI-BD-TPB) to generate hydrogen peroxide (H₂O₂). The photoreduction of the carbonyl group into anion radical intermediate promotes the adsorption of O₂, thereby boosting the catalytic performance. While the concept of this work is interesting, reasonable explanations of the reaction mechanism and experimental results are lacking. More data and relative discussion should be added to make the conclusion more convincing.

Response: We thank you very much for your valuable suggestion.

1. Comments: This article reported that both electron reduction and hole oxidation produce H₂O₂. What is the stoichiometric ratio of H₂O₂ produced by oxidation and reduction?

Response: Thank you for your professional comment. Following your guidance, we have supplemented the related data to study the contribution of electron and hole. We have evaluated the stoichiometric ratio of H₂O₂ produced via the oxidation and reduction reaction by isotopic experiments. The stoichiometric ratio of H₂O₂ produced via the oxidation and reduction reaction is close to 1:1.

We design four sets of isotope experiments:

#1 as a blank control: the system of H₂¹⁶O and ¹⁸O₂, containing PI-BD-TPB polyimide aerogel photocatalyst; light source: dark; (#1 and #2 are controls for each other)

#2 as a control: the system of H₂¹⁶O and ¹⁸O₂, containing PI-BD-TPB polyimide aerogel photocatalyst; light source: 300 W Xe lamp with as the simulated sunlight;

#3 as a blank control: the system of H₂¹⁸O and ¹⁶O₂, containing PI-BD-TPB polyimide aerogel photocatalyst; light source: dark; (#3 and #4 are controls for each other)

#4 as a control: the system of H₂¹⁸O and ¹⁶O₂, containing PI-BD-TPB polyimide aerogel photocatalyst; light source: 300 W Xe lamp with as the simulated sunlight;

Table S2 Isotopic experiments with ¹⁸O₂ or H₂¹⁸O for H₂O₂ production on PI-BD-TPB photocatalyst after reaction.

Entry	O ₂ source	H ₂ O source	Dark or Light	nH ₂ O ₂ mmol	nH ₂ ¹⁸ O ₂ mmol	n ¹⁸ O/n ¹⁶ O ppm
#1	¹⁸ O ₂	H ₂ ¹⁶ O	Dark	0	0	1988.1279
#2	¹⁸ O ₂	H ₂ ¹⁶ O	Light	0.1269	0.0652	2184.0358
#3	¹⁶ O ₂	H ₂ ¹⁸ O	Dark	0	0	1992.7183
#4	¹⁶ O ₂	H ₂ ¹⁸ O	Light	0.1040	0.0512	2146.3579

The reaction was performed in a sealed reactor with first evacuated and then refilled with ¹⁸O₂ (97% atom ¹⁸O purity) or H₂¹⁸O (97% atom ¹⁸O purity). The peroxy species

are detected by converting them to water. First, potassium titanium oxalate was added into the solution (after reaction) to form peroxy complex, and dried via frozen drying. Then the collected potassium titanium oxalate-H₂O₂ complex (for #1 and #2). #3 and #4 are re-dissolved and dried again (five times) to remove possible adsorbed H₂¹⁸O on the PI-BD-TPB surface. The potassium titanium oxalate-H₂O₂ complex was reduced with 12 mL KI (aq), in order to convert the contained H₂O₂ to H₂O. The H₂O including that is converted from the H₂O₂ is collected by distillation and detected by Liquid Water Isotope Analyzer. Finally, the ratio of n¹⁸O / n¹⁶O obtained directly from the Liquid Water Isotope Analyzer. The amount of ¹⁸O from water in #1 and #3 is calculated as follows:

Eq. 1..... $m(\text{H}_2\text{O}) = n^{18}\text{O}(\text{H}_2\text{O}) / n^{16}\text{O}(\text{H}_2\text{O})$

Eq. 2..... $n\text{O}(\text{H}_2\text{O}) = n^{16}\text{O}(\text{H}_2\text{O}) + n^{18}\text{O}(\text{H}_2\text{O})$

where n¹⁸O(H₂O) is the ratio of ¹⁸O in the H₂O; n¹⁶O(H₂O) is the ratio of ¹⁶O in the H₂O; m(H₂O) is the ratio between n¹⁸O(H₂O) and n¹⁶O(H₂O) (directly obtained from the measurements); and nO (H₂O) is the total amount of O in the H₂O. nO(H₂O) is 665.9 mmol in 12 mL KI solution. The average abundances of the oxygen isotopes ¹⁶O and ¹⁸O in nature are 99.762% and 0.200%, respectively.

By combining **Eq. 1** and **Eq. 2**, n¹⁸O (H₂O) and n¹⁶O (H₂O) can be calculated as:

Eq. 3..... $n^{18}\text{O}(\text{H}_2\text{O}) = (m(\text{H}_2\text{O}) \times n\text{O}) / (1 + m(\text{H}_2\text{O}))$

Eq. 4..... $n^{16}\text{O}(\text{H}_2\text{O}) = n\text{O} / (1 + m(\text{H}_2\text{O}))$

Accordingly, n¹⁸O (H₂O) and n¹⁶O (H₂O) is calculated as 1.321 mmol and 664.7 mmol, respectively, in #1, where m (H₂O) is 1988.1279 ppm. And n¹⁸O (H₂O) and n¹⁶O (H₂O) is calculated as 1.324 mmol and 664.7 mmol, respectively. In #3, where m (H₂O) is 1992.7183 ppm. The amount of ¹⁸O in H₂O₂ in #2 and #4 is calculated based on the results in #1 and #3, concurrently.

Eq. 5..... $m = [n^{18}\text{O}(\text{H}_2\text{O}_2) + n^{18}\text{O}(\text{H}_2\text{O})] / [n^{16}\text{O}(\text{H}_2\text{O}_2) + n^{16}\text{O}(\text{H}_2\text{O})]$

Eq. 6..... $n\text{O}(\text{H}_2\text{O}_2) = n^{16}\text{O}(\text{H}_2\text{O}_2) + n^{18}\text{O}(\text{H}_2\text{O}_2)$

By combining **Eq. 5** and **Eq. 6**, n¹⁸O (H₂O₂) can be calculated as:

Eq. 7..... $n^{18}\text{O}(\text{H}_2\text{O}_2) = (m \times (n^{16}\text{O}(\text{H}_2\text{O}) + n\text{O}) - n^{18}\text{O}) / (1 + m)$

Therefore, for #2 (¹⁸O₂ + H₂¹⁶O), the amount of O in H₂O₂ (0.2539 mmol, C_{H2O2} = 6.3466 mmol/L, V_{aq} = 20 mL) is calculated based on the concentration of H₂O₂, and m is measured to be 2184.0358 ppm. Then n¹⁸O (H₂O₂) is calculated to be 0.1304 mmol, and thus the amount of H₂¹⁸O₂ is 0.0652 mmol. The total H₂O₂ production amount is measured to be 0.1269 mmol. Therefore, half of the H₂O₂ is mainly produced via the electron reduction reaction.

For #4 ($^{16}\text{O}_2 + \text{H}_2^{18}\text{O}$), the amount of O in H_2O_2 (0.2080 mmol, $C_{\text{H}_2\text{O}_2} = 5.2006 \text{ mmol/L}$, $V_{\text{aq}} = 20 \text{ mL}$) is calculated based on the concentration of H_2O_2 , and m is measured to be 2146.3579 ppm. Then $n^{18}\text{O}(\text{H}_2\text{O}_2)$ is calculated to be 0.1023 mmol, and the amount of $\text{H}_2^{18}\text{O}_2$ is 0.0512 mmol. The total H_2O_2 production amount is measured to be 0.1040 mmol. Therefore, near half of the H_2O_2 is mainly produced via the hole oxidation reaction.

Isotopic experiments with $^{18}\text{O}_2$ or H_2^{18}O for H_2O_2 production on the PI-BD-TPB photocatalyst.

2.Comments: *The EPR signal in Fig. 2f was described as imide radical. Why? This EPR signal may also correspond to defects in the catalyst.*

Response: Thank you for your meticulous comment, we would like to explain this issue in detail. The electron paramagnetic resonances (EPR) testing was used to monitor defects in the catalyst. According to relevant literature, the g value of 2.004 or 2.009 or 2.0071 or 2.0082 was identified as the electrons trapped on defects in the catalyst (*J. Am. Chem. Soc.* **2020**, 142, 4298; *J. Am. Chem. Soc.* **2014**, 136,6826; *Nature communications* **2021**, 12, 2039.). However, there are no defects in PI-BD-TPB aerogel photocatalyst, due to the results of FTIR (**Fig. 2b**), solid-state ^{13}C NMR (**Fig. 2c**) and XPS spectra (**Supplementary Figure S17**). Furthermore, catalyst with defects also can exhibit an ESR signal at $g = 2.00342$, which was attributable to the unpaired electrons in the π -conjugated unit (*Applied Catalysis B: Environmental* **2020**, 302, 120845). The ESR signal $g = 2.003$ in the dark is used to compare the degree of defect in different catalyst. Our experimental environment was light off and then light on, in order to compare the signal intensity of polyimide aerogel photocatalyst before and after light

irradiation. There was no obvious signal at $g = 2.003$ in the dark condition. Upon light irradiation, the intensity of signal at $g = 2.003$ strengthened significantly, showing that photo-induced electrons transferred to carbonyl groups. We have proved in the revised supporting information that carbonyl groups have strong electron affinity characteristic and easily accept photogenerated electrons under photoexcitation. With the extension of illumination time, the signal intensity is enhanced, showing that more and more photo-induced electrons transfer to carbonyl groups. The photogenerated electrons delocalized near the carbonyl groups and formed anion intermediates due to the conjugated structure. Combined with the results of *in situ* XPS spectra (**Fig. 2e**), the anion radical is attributed to the imide radical, consistent with the literature reports (*Adv Sci.* **2023**, 10, e2301017; *Chem. Sci.* **2015**, 6, 3975; *Angew. Chem.* **2015**, 54, 3611; *J. Am. Chem. Soc.* **2007**, 129, 6354; *Journal of Energy Chemistry* **2022**, 69 428; *J. Am. Chem. Soc.* **2012**, 134, 13679; *Mater. Chem. Front.*, **2022**, 6, 2545; *Angew. Chem.* **2022**, 61, e202207221).

3.Comments: The authors said that photogenerated-hole can oxidize water to produce H_2O_2 . What active species are involved during this process? Please provide experimental evidence.

Response: We greatly appreciate your detailed comments, we would like to explain in detail here. Hydroxyl radical specie was not involved during water oxidation to H_2O_2 process in our system. The explanation of water oxidation was not mentioned in the previous manuscript, in fact, the relevant experimental evidences were in the previous supporting information. For the PI-BD-TPB photocatalyst, the VB position at 2.02 V (vs. RHE) indicated that photogenerated holes were not able to oxide water to form hydroxide radical (**Supplementary Figure S42**). Hydroxyl radical is captured in 5,5-dimethyl-1-pyrroline N-oxide (DMPO) and H_2O . In dark, no EPR signals were observed, while hydroxide radical was also not observed for BD-TPB after irradiating 5 min (**Supplementary Figure S52**). Meanwhile, in the presence of tert-butyl alcohol (t-BuOH) served as hydroxyl radical scavenger, the concentration of H_2O_2 production was not decreased significantly, showing that hydroxyl radical was not key active specie (**Supplementary Figure S50**). Therefore, we confirmed that H_2O_2 on PI-BD-TPB photocatalyst was generated via a directly two-electron water oxidation pathway. For the two-electron water oxidation pathway, the largest Gibbs free energy step is the first oxidation step to form OH^* . The formation of OH^* is crucial for the generation of H_2O_2 . The adsorption energy of OH^* on the Site 3 (1.52 eV) was lowest (**Supplementary**

Figure S54b). The water oxidation reaction most likely occurred at C atom in TPB unit. Therefore, hydroxyl radical specie was not involved during H₂O₂ production. The water oxidation site was on C atom from TPB unit via a directly two-electron water oxidation pathway for H₂O₂ production.

Electron paramagnetic resonance spectral for capturing hydroxyl radical of the BD-TPB.

The photocatalytic H₂O₂ concentration of the BD-TPB under different reaction condition.

(a) Calculated free energy diagrams of two-electron water oxidization pathway toward H₂O₂ production on different active sites; (b) The adsorption energy of OH* on different sites of the BD-TPB.

4.Comments: Which specific anion radical does the signal at 418 nm correspond to (Fig. 2g)? Further experimental evidence and detailed structural formulas should be given. In addition, the role of Na₂S₂O₄ also needs to be explained in detail.

Response: Thank you for your professional suggestions, we would like to explain in detail from many aspects. The change absorption in UV-vis spectra is only one of evidences to confirm the formation of imide anion radical. And we combine fluorescence test, *in situ* XPS, ESR test and related literatures together to illustrate the formation and structure of the imide anion radical.

- a) The imide anion radical is essentially a type of carbon-oxygen anion radical. Carbonyl groups with strong electrophilicity easily accept photogenerated electrons in photoexcitation and further form the anion radical. According to the relevant literature (*J. Phys. Chem. A* **2000**, 104, 6545-6551; *J. Am. Chem. Soc.* **2007**, 129, 20, 6354-6355; *J. Am. Chem. Soc.* **2020**, 142, 2204-2207; *Science*, **2014**, 346(6210):725-728.), UV-vis absorption spectra and fluorescence spectra combined with electron paramagnetic resonance test are collectively effective means of probing the formation of imide anion radical.
- b) In the UV-vis absorption spectrum (**Fig. 2g**), the 250-300 nm absorption peak was from carbonyl groups on the imide ring. Upon illumination, carbonyl group from imide ring received many electrons provided by Na₂S₂O₄ as the electron donor, the structure changed, and the absorption appeared red-shifted. The new peak at 418 nm corresponded to the carbon-oxygen anion radical from by the carbonyl group receiving photogenerated electrons (*Macromolecules* **1999**, 32, 387-396; *Chem. Sci.*, **2015**, 6, 3975).

Ultraviolet-visible absorption spectra of BD-TPB and BD-TPB* anion radical in DMF solution with Na₂S₂O₄ as electron donor under Ar atmosphere in dark and light condition.

- c) The formation of the anion radical changes the fluorescence intensity of PI-BD-TPB photocatalyst. In order to confirm this, we performed fluorescence test (**Supplementary Figure S49**). Under the dark condition, PI-BD-TPB exhibited an obvious signal at 460 nm. The signal at 460 nm decreased after light irradiation, indicating that the carbonyl group received photogenerated electrons and formed the anion radical, leading to a decrease in the fluorescence signal (*Chem. Sci.*, **2015**, 6, 3975; *J. Phys. Chem. A*, **2009**, 113, 1747; *Langmuir*, **2007**, 23, 11972; *J. Am. Chem. Soc.* **2012**, 134, 386).

Fluorescence spectrum of BD-TPB and BD-TPB* anion radical in DMF solution with $\text{Na}_2\text{S}_2\text{O}_4$ as electron donor under Ar atmosphere in dark and light condition. (Excitation wavelength at 365 nm; emission wavelength at 460 nm).

- d) Meanwhile, electron paramagnetic resonance spectroscopy was performed on the PI-BD-TPB to confirm the changes in carbonyl groups (**Fig. 2f**). Compared to signals generated in the dark, a distinct signal appeared at $g = 2.003$ after illumination and strengthened with light time. The formed signal corresponds to the imide anion radical according to literature reports (*Adv Sci.* **2023**, 10, e2301017; *Chem. Sci.*, **2015**, 6, 3975; *Angew. Chem., Int. Ed.* **2015**, 54, 3611; *J. Am. Chem. Soc.* **2007**, 129, 6354; *Journal of Energy Chemistry* **2022**, 69 428; *J. Am. Chem. Soc.* **2012**, 134, 13679; *Mater. Chem. Front.*, **2022**, 6, 2545; *Angew. Chem. Int. Ed.* **2022**, 61, e202207221). These changes demonstrated that anion radical produced on PI-BD-TPB surface during light illumination process.
- e) Besides, *in situ* X-ray photoelectron spectroscopy also was monitored the structural variation of imide anion radical (**Fig. 2e**). In the initial dark, the O 1s XPS peak was resolved into one major component (532.1 eV) for the C=O group. Under light irradiation, a prominent new peak at 530.8 eV attributed to the C–O bond emerged (*Angew. Chem.* **2022**, 61, e202117661; *Angew. Chem.* **2022**, 61, e202207043). This

observation showed that the C=O group on imide ring received electrons to form C–O group.

Therefore, combined UV-vis absorption spectrum, fluorescence test, *in situ* XPS, ESR test and related literatures, the schematic representation of the imide anion radical in our system is shown below.

Electron paramagnetic resonance spectra of the PI-BD-TPB before and after light irradiation.

In situ X-ray photoelectron spectroscopy spectrum of O 1s of PI-BD-TPB before and after light.

5.Comments: Supplementary Fig.15-36 should be added to the manuscript with reasonable statements and explanations.

Response: Based on your detailed suggestion, we have provided the detailed description of this section in the revised manuscript. The specific additions in the revised manuscript are as follows:

Additional characterizations were carefully performed to understand the features of the PI-BD-TPB aerogel. PAA is the precursor of polyimide, whose molecular weight distribution determines the molecular weight of the final product. Due to the insolubility of PI-BD-TPB, we measured the molecular weight of the PAA gel powders via gel permeation chromatography in NMP. The average molecular weight (Mw) and corresponding polydispersity index (PDI, $PDI = M_w/M_n$, $M_n = 3.4$ kDa) were 8.0 kDa and 2.35 (**Supplementary Figs. 18**). SEM and TEM images of the PI-BD-TPB aerogel (**Supplementary Figs. 19-21**) revealed cross-linked spherical particles with an average diameter of ~ 1 μm . High-resolution transmission electron microscopy (HRTEM) images revealed that PI-BD-TPB was locally crystalline in nature. The interlayer distance was measured to be 0.43 nm (**Supplementary Figs. 22**). Furthermore, the XRD peak of PI-BD-TPB was located at $2\theta \approx 20.3^\circ$, representing the 0.43 nm interlamellar d-spacing of π - π stacking (**Supplementary Figs. 23**). The porosity of PI-BD-TPB was assessed using N_2 sorption measurements at 77.3 K, and the Brunauer–Emmett–Teller (BET) surface area was calculated to be $372.8 \text{ m}^2 \text{ g}^{-1}$. By employing a nonlocal density functional theory (NLDFT) model, its pore size was determined to be ~ 1.5 nm (**Supplementary Figs. 24**). The macroscopic pore size distribution on the surface of the BD-TPB aerogel was analyzed by mercury intrusion porosimetry (**Supplementary Figs. 25**). The macroscopic pore size on the surface of BD-TPB was ~ 2 μm .

Thermogravimetric analysis revealed that the PI-BD-TPB aerogel had a high thermal stability up to 550°C (**Supplementary Figs. 26**). The chemical stability of the PI-BD-TPB aerogel was investigated by immersing it in diverse solvents. Notably, the FTIR spectra of PI-BD-TPB after soaking in different solvents were almost unchanged, confirming its excellent chemical stability (**Supplementary Figs. 27**). The outstanding chemical and thermal stability are ascribed to the strong imide linkage and highly conjugated structure. The average ζ potential of PI-BD-TPB was -55.1 mV in H_2O aqueous solution, indicating strong adsorption of H^+ (**Supplementary Figs. 28**). PI-BD-TPB displayed a hydrophilic surface with a contact angle of 40.4° (**Supplementary Figs. 29**). The macroscopic polyimide aerogel was cylindrical, with a diameter of 3 cm and a height of 2 cm. Compared to powder with the same weight, it was characterized by a low density (ca. 37.78 mg/cm^3). (**Supplementary Figs. 30-32**).

Adsorption is a special feature of aerogel materials. The 0.53 g PI-BD-TPB aerogel rapidly absorbed 13.89 g of H_2O after 5 min, which was approximately 25 times its weight, exhibiting an effective adsorption capacity (**Supplementary Figs. 33 and 34**).

In addition, the PI-BD-TPB aerogel with outstanding mechanical behavior could hold 200 times its own weight (**Supplementary Figs. 35**). A compression–recovery test was performed to estimate the mechanical durability, and the PI-BD-TPB aerogel, as an elastic material, exhibited excellent reversible compressibility at strains of 10%, 20%, 30%, 40%, 50% and 60% (**Supplementary Figs. 36**). Benefiting from the facile synthetic route, a macroscopic polyimide membrane was also prepared, further exhibiting high operability (**Supplementary Figs. 37**). In summary, the covalently crosslinked polyimide aerogel has a low density, hydrophilicity, an effective absorption ability, high chemical and thermal stability, excellent resilience and good mechanical behavior.

Theoretical calculations showed that the highest occupied molecular orbital (HOMO) was mainly distributed on the TPB unit, while the lowest unoccupied molecular orbital (LUMO) was distributed on the BD unit, suggesting transfer of photogenerated electrons from the TPB unit to the BD unit (**Supplementary Figs. 38**). The -C=O groups on BD-TPB had electron affinity characteristics according to the electrostatic potential distribution (**Supplementary Figs. 39**).

6. Comments: Is this anion radical intermediate-mediated H_2O_2 synthesis strategy universal? Please test at least five other photocatalysts with carbonyl groups for evaluation.

Response: We are grateful for your thoughtful review of our manuscript. We think that anion radical intermediate-mediated H_2O_2 synthesis strategy is universal to some extent. Our strategy has a certain degree of universality under certain conditions such as carbonyl group from imide ring, cross-linked frameworks, hyper-conjugated donor units.

- (1) Robust imide linkage has conjugated property, which is conducive to charge transport to carbonyl group.
- (2) Cross-linked frameworks facilitate charge separation and transfer in multiple directions, providing sufficient electrons to carbonyl group, which drives the formation of anion radical intermediate.
- (3) The hyper-conjugated structure is conducive to the stable formation of anion radical intermediates.

To verify the above results, we design three classes of carbonyl-containing photocatalysts: a) carbonyl groups on non-imide ring; d), non-crosslinked structure; c) non-conjugated donor unit. The key experiment to verify the mechanism of anion

radical intermediate-mediated H₂O₂ synthesis is that H₂O₂ can be produced in the dark reaction. Nine photocatalysts with carbonyl group were prepared and evaluated, respectively. We preliminarily confirmed the structure of nine photocatalysts by FTIR spectroscopy. H₂O₂ was produced on nine photocatalysts with carbonyl group in 5 mM vitamin C solution as electron donor in argon atmosphere. Oxygen quickly was kept bubbling to the closed system for 30 min after light irradiation for 60 min. The H₂O₂ concentration was determined by potassium titanium oxalate method.

- a) The synthesis of #A PAA photocatalyst was described in the supporting information. No H₂O₂ was detected by PAA photocatalyst under “Dark-O₂” condition, showing that imide ring played a crucial part in anion radical intermediate-mediated H₂O₂ production. The carbonyl group from imide ring is the key prerequisite for anion radical intermediate generation.
- b) Three linear polyimide (#B1-3) were prepared by the imidization of aromatic triamine with aromatic dianhydride. Briefly, benzene-1,4-diamine (8.1 mg, 0.75 mmol) and BD (22.1 mg, 0.75 mmol) were dispersed into the mixture of 2.1 mL NMP /Mesitylene/ Isoquinoline (1: 1: 0.1 = v: v: v), respectively. After being sonication with the power of 100 W for 20 min, and then the polymerization reaction was sealed for 48 h at 180 °C. The precipitate was washed with the mixture of NMP /ethanol. The product was dried under vacuum to obtain linear polyimide powers named as B1-LP. The B2-LP and B3-LP were synthesized under the same experimental procedure. (Benzidine (13.8 mg, 0.75 mmol) and [1,1':4,1''-terphenyl]-4,4''-diamine (19.5 mg, 0.75 mmol)). Similarly, no hydrogen peroxide production under “Dark-O₂” condition was detected on three linear polyimide (B1-LP, B2-LP and B3-LP). This result suggests that crosslinked framework structures also play an important role in the formation of anion radical intermediate.
- c) #C1-2 photocatalyst were also prepared by the imidization of aromatic triamine with aromatic dianhydride. Typically, BD (53 mg, 0.9 mmol) dissolved in 2 mL of 1-methyl-2-pyrrolidinone (NMP) solution, then 1,3,5-triaminobenzene (TAB) or 1,3,5-tris(4-aminophenyl) benzene (TBZ) (0.6mmol) dispersed in 2 mL of Mesitylene solution, then Isoquinoline (0.2 mL) was added. The mixture was upon further solvothermal treatment at 160 °C for 48 h. The products were washed in a solution of 75% NMP in ethanol for 24 h. Subsequently, the solvent was exchanged for three times. Finally, #C1 and #C2 photocatalyst was obtained by freeze-dried. Interestingly, we detected hydrogen peroxide production from #C2 photocatalyst after “Dark-O₂” reaction, but no hydrogen peroxide was produced in #C1

photocatalyst. We further speculate that hyper-conjugated donor unit not only promotes charge separation but also facilitates stabilization of anion radical intermediate.

- d) To verify the above-mentioned hypothesis, we designed and synthesized **#D** photocatalyst (DA-TABPB) and **#E** photocatalyst (BD-Py) with hyper-conjugated donor unit. Hydrogen peroxide production was also detected after “Dark-O₂” reaction in both **#D1** and **#E1** photocatalyst.

#A photocatalyst with carbonyl groups (PAA photocatalyst)

Structure diagram (right) and FTIR spectra (left) of the PAA photocatalyst.

#B1-3 photocatalyst with carbonyl groups (LP photocatalyst)

Structure diagram (up) and FTIR (on) spectra of linear polymer photocatalysts.

#C1-2 photocatalyst with carbonyl groups (CP photocatalyst)

Structure diagram (up) and FTIR (on) spectra of cross-linked polymer photocatalysts.

#D photocatalyst with carbonyl groups (DA-TABPB photocatalyst)

Structure diagram (right) and FTIR spectra (left) of DA-TABPB photocatalyst.

#E photocatalyst with carbonyl groups (HCP photocatalyst)

Structure diagram (right) and FTIR spectra (left) of hyper-crosslinked polymer photocatalyst.

H₂O₂ production in 5 mM vitamin C as electron donor solution in argon atmosphere on nine photocatalysts with carbonyl group under “Dark-O₂” condition. Reaction conditions: 20 mg photocatalyst, 15 mL solution, 300 W Xe lamp.

7.Comments: Under actual electrochemical reaction conditions, the surface charge of the catalyst varies with the electrode potential. Therefore, a more appropriate calculation method (constant potential approach) and solvation models should be used to describe the real electrochemical behavior during the synthesis of H₂O₂.

Response: According to your detailed advice, we have supplemented the

electrochemical performance for H₂O₂ generation on PI-BD-TPB photocatalyst via constant potential method using different electrolyte solutions. We measured the concentration of H₂O₂ and calculated the corresponding faradic efficiency (FE) in the flow cell device with O₂-saturated 0.5 M Na₂SO₄ solution as neutral electrolyte or 0.5 M H₂SO₄ solution as acidic electrolyte (*Energy Environ. Sci.* **2022**,15, 4167). We used PI-BD-TPB/carbon paper (1.0 cm²) as working electrode, Ag/AgCl electrode as the reference electrode and platinum wire electrode as the counter electrode, respectively. 0.5 M Na₂SO₄ solution and 0.5 M H₂SO₄ solution were employed as neutral and acidic electrolyte for O₂ reduction reaction, respectively. Hydrogen peroxide was formed in the electrolyte solution. We took the electrolyte solution and determined the concentration of H₂O₂ by potassium titanium oxalate method. The FE for H₂O₂ production is calculated according to the following equation:

$$FE(\%) = \frac{N \times F \times n_{H_2O_2}}{Q} \times 100\% \dots \dots \dots$$

where N is the number of charge transfer for H₂O₂ production (“2” in the reaction), F is the Faraday constant (96485 C mol⁻¹), n_{H₂O₂} is the produced H₂O₂ (mol), and Q (C) is the charge transfer number for the whole reaction, which can be obtained from the I-t curves.

The linear sweep voltammetry curves in **Supplementary Figure S66a** and **S66c** exhibited that the currents of PI-BD-TPB catalyst recorded in the O₂-saturated with 0.5 M Na₂SO₄ solution or 0.5 M H₂SO₄ solution, respectively. Due to the significant oxygen reduction current at 0.2 V vs. RHE in 0.5 M Na₂SO₄ neutral electrolyte, constant potential tests were carried out at 0.2 V, 0.0 V, -0.2 V and -0.4 V vs. RHE applied potentials. In neutral electrolyte, the H₂O₂ concentration at the cathode side was measured to be 3.09 mM, 12.49 mM, 17.21 mM and 27.33 mM at the corresponding potentials of 0.2 V, 0.0 V, -0.2 V and -0.4 V vs. RHE, respectively. Therefore, the corresponding FE% was 72.41%, 90.94%, 82.84% and 91.72%, respectively (**Supplementary Figure S66b**). Similarly, constant potential tests were carried out at -0.3 V, -0.5 V, -0.7 V and -0.8 V vs. RHE potentials in 0.5 M H₂SO₄ acidic electrolyte due to the significant oxygen reduction current at -0.3 V vs. RHE. The H₂O₂ concentration at the cathode side was measured to be 6.91 mM, 20.23 mM, 41.95 mM and 61.67 mM in acidic electrolyte at the corresponding potentials of -0.3 V, -0.5 V, -0.7 V and -0.8 V vs. RHE, respectively. The FE was calculated to 70.76%, 63.22%, 58.32% and 61.04% at the corresponding potential (**Supplementary Figure S66d**). Therefore, the above results show that the PI-BD-TPB photocatalyst has excellent two-electron oxygen reduction for H₂O₂ electrosynthesis, excluding the possibility of the

surface charge of the catalyst varies with the electrode potential. This demonstrates the accuracy of test results involving the electrochemical synthesis of H_2O_2 (Supplementary Figure S53, S64 and S65).

Electrocatalytic performance for the PI-BD-TPB catalyst toward two-electron oxygen reduction for H_2O_2 electrosynthesis in neutral and acidic electrolyte. (a) The linear sweep voltammetry curve of PI-BD-TPB catalyst obtained under 50 mV s^{-1} in O_2 -saturated $0.5 \text{ M Na}_2\text{SO}_4$; (b) H_2O_2 concentration and the corresponding faradic efficiency (FE) % of PI-BD-TPB catalyst at different applied potentials in the flow cell device with O_2 -saturated $0.5 \text{ M Na}_2\text{SO}_4$; (c) LSV curve of PI-BD-TPB catalyst obtained under 50 mV s^{-1} in O_2 -saturated $0.5 \text{ M H}_2\text{SO}_4$; (d) H_2O_2 concentration and the corresponding FE % of PI-BD-TPB catalyst at different applied potentials in the flow cell device with O_2 -saturated $0.5 \text{ M H}_2\text{SO}_4$.

REVIEWERS' COMMENTS

Reviewer #1 (Remarks to the Author):

The revised manuscript has been carefully reviewed. The reviewer is satisfied with all the revisions made. This revised manuscript is recommended for publication as it is. Congratulations!

Reviewer #2 (Remarks to the Author):

The revision addressed my comments well and it can be accepted as it is.

Reviewer #3 (Remarks to the Author):

The authors has carefully address my concerns. The manuscript can be published as is.

Response to Reviewers

Manuscript Title: A photocatalytic redox cycle over a polyimide catalyst drives efficient solar-to-H₂O₂ conversion

Manuscript Number: NCOMMS-24-03300B

Author: Wenwen Chi⁺, Yuming Dong^{+*} and Yongfa Zhu^{*}

Reviewer #1 (Remarks to the Author):

The revised manuscript has been carefully reviewed. The reviewer is satisfied with all the revisions made. This revised manuscript is recommended for publication as it is. Congratulations!

Response: We are thankful for the Reviewer's helps and positive evaluation of our manuscript.

Reviewer #2 (Remarks to the Author):

The revision addressed my comments well and it can be accepted as it is.

Response: We are thankful for the Reviewer's helps and positive evaluation of our manuscript.

Reviewer #3 (Remarks to the Author):

The authors has carefully address my concerns. The manuscript can be published as is.

Response: We are very thankful for the Reviewer's positive evaluation of our work.